# Observing the tidal pulse of rivers from wide-swath satellite altimetry

M. G. Hart-Davis[1,7] ✉, D. Scherer[1,7], C. Schwatke[1], A. H. Sawyer[2,3], T. M. Pavelsky[4], R. D. Ray[5], P. Matte[6], D. Dettmering[1] & F. Seitz[1]

The characteristic tides of coastal rivers influence the distribution of estuarine and wetland habitats[1], the extent of fresh drinking water[2], carbon and nitrogen cycles[3,4], and sediment export to the ocean[5]. Despite the importance of riverine tides, their range is generally unknown over most of the world's rivers because the propagation of tidal waves in channels is complex, gauging stations are scarce and conventional nadir altimetry[6] has historically been too sparse for use in rivers. Here we use data from the recently launched Surface Water and Ocean Topography (SWOT) satellite to quantify tidal dynamics across 3,172 coastal rivers. Capitalizing on the wide swath coverage of SWOT, we show that over 165,000 river kilometres are influenced by tides. More than 700 million people live near and depend on these coastal transition zones. River size, slope and tidal range at the river mouth influence the extent to which tides propagate within river systems. Natural and artificial obstacles, such as dams, limit tidal propagation in an estimated 16% of all tidal rivers. The tidal dataset opens new possibilities for monitoring and modelling changes in estuarine habitats, fresh drinking water for coastal cities and riverine carbon budgets[7] across annual to decadal timescales in response to sea-level rise[8], megadrought[9], intensifying water extraction and river regulation[10].

The importance of tides is embodied in the origins of the word 'estuary', which comes from the Latin word 'aestuarium', meaning the place of the tide. Along coastlines, where tides are typically magnified[11], they profoundly affect navigation, commerce, coastal flooding, water properties and sediment transport[12]. Tides impact the flooding of rivers and, thus, influence the extent of their floodplain, which has cascading effects on biogeochemical and ecological processes[13]. Flood cycles, modulated by tides and sea-level rises, are a key factor in the dieback or persistence of saltmarsh systems[14]. Tidal flooding also accelerates both nitrification and denitrification in soils, leading to variable effects on nutrient levels and coastal water quality[15,16]. Tides enhance the mixing of saline and freshwater along river channels, which affects water security in coastal cities and food security for agricultural deltas that use river water for irrigation[17]. Moreover, tides are a critical factor in compound coastal flooding, as demonstrated by the devastating combination of storm surge and high tide that hit New York City during Hurricane Sandy in 2012[18]. Despite their importance, the extent of tides in coastal rivers is poorly known on a global scale because tidal propagation depends on the unique morphologic and hydrologic conditions of each river as well as the magnitude of the tidal signal at the river mouth[19]. The relative importance of these controls on tidal extent—indeed, the extent itself—has not been unravelled on a global scale, primarily due to limitations of observational coverage.

The combination of numerical modelling and satellite altimetry[20] now gives us the ability to predict tidal elevations anywhere in the open ocean with accuracies approaching 1 cm (ref. 21). However, coastal regions and connected inland water bodies remain the Achilles heel for tidal prediction[6]. The challenge largely stems from two constraints of traditional nadir altimetry: land contamination of radar returns and inadequate density of satellite overpasses. Moreover, nonlinear effects of the shallow water in rivers and estuaries drive complex tidal behaviours[22,23], which are best described with physics-based models. Yet adequately resolved models exist only for a handful of locations, owing to the scarcity of bathymetric and water-level data to constrain models and the complexity of the dynamics, which vary from river to river[19,24,25].

The highly anticipated Surface Water and Ocean Topography (SWOT) satellite provides two-dimensional swath observations of both ocean and inland water surfaces[26]. SWOT has the potential to revolutionize the scientific understanding of water-related physical processes[27] and to open new avenues of scientific research across the land–ocean–aquatic continuum[28,29]. Early studies have shown that SWOT can improve the accuracy of tidal estimates in coastal regions, and they have hinted at the potential of using these data to study tidal processes within inland water bodies, including rivers[30,31].

The question then becomes, can SWOT be used to pioneer a deeper understanding of tidal dynamics within coastal rivers worldwide? If so, could these insights support the creation of the first-ever global atlas of tidal extent within rivers? By exploiting the observational capabilities of SWOT at the land–ocean interface, we address both these questions and produce a global atlas of tidal rivers, which could

[1]Deutsches Geodätisches Forschungsinstitut, Technische Universität München, Munich, Germany. [2]Department of Civil and Environmental Engineering (DECA), Universitat Politècnica de Catalunya (UPC), Barcelona, Spain. [3]Associated Unit: Hydrogeology Group (UPC-CSIC), Barcelona, Spain. [4]University of North Carolina, Chapel Hill, NC, USA. [5]Geodesy & Geophysics Lab., NASA Goddard Space Flight Center, Greenbelt, MD, USA. [6]Meteorological Research Division, Environment and Climate Change Canada, Quebec City, Quebec, Canada. [7]These authors contributed equally: M. G. Hart-Davis, D. Scherer. ✉e-mail: michael.hart-davis@tum.de

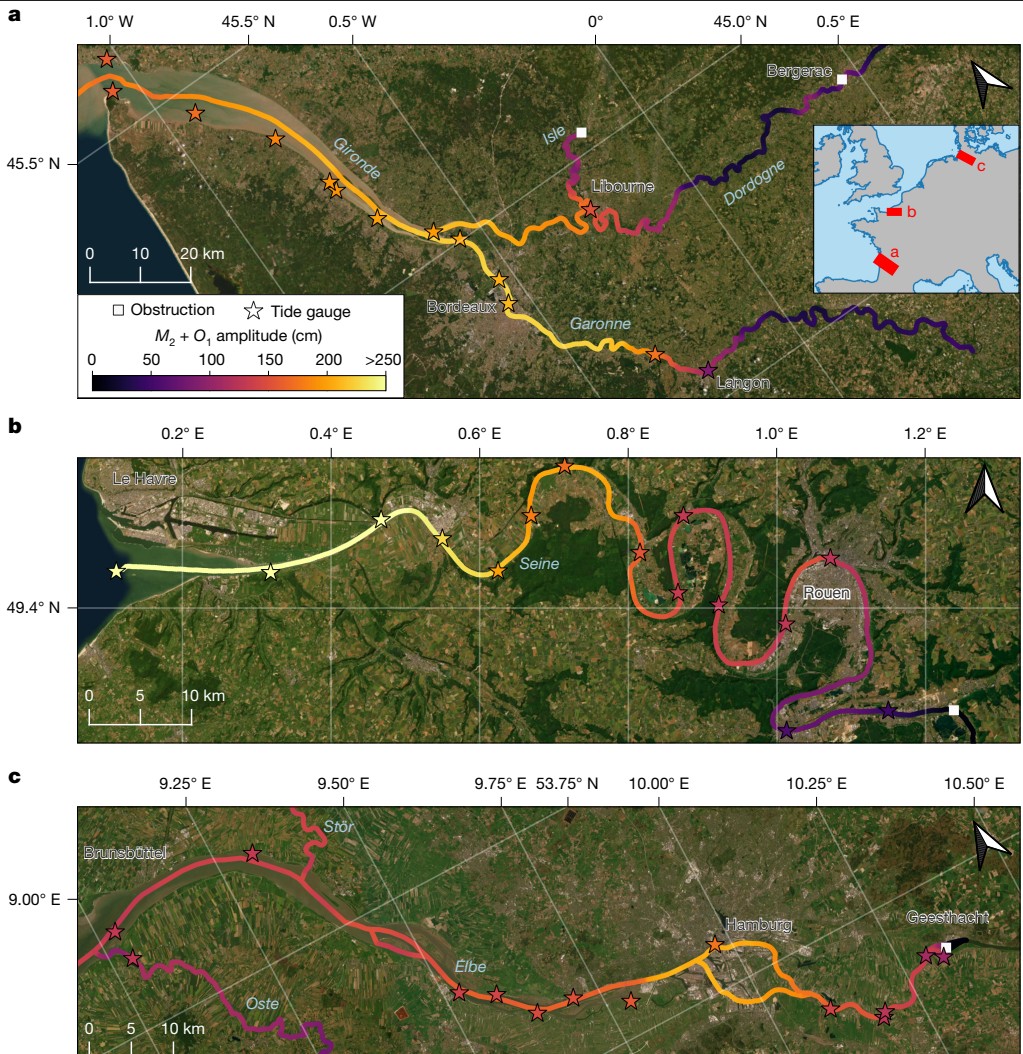

**Fig. 1 | Summed $M_2$ and $O_1$ tidal amplitudes within three selected river networks, estimated from SWOT elevation measurements. a–c,** The three selected regions are the Gironde Estuary (**a**), the Seine River (**b**) and the Elbe River (**c**). Stars with black borders along the trajectory of the rivers represent the summed $M_2$ and $O_1$ amplitude calculated from in situ river gauge observations. The white squares mark the location of dams. Satellite base maps powered by Esri.

radically expand our understanding of the fundamental tidal force that controls land–ocean interactions.

## Observing river tides from space

To assess the feasibility of using SWOT data to estimate tides within rivers, we conducted a harmonic analysis on the SWOT River Single-Pass Vector Data Product (RiverSP) from March 2023 to May 2025 to estimate the diurnal $O_1$ and semi-diurnal $M_2$ tidal constituents for all observed rivers. Global amplitude results are available at the Database for Hydrological Time Series of Inland Waters (DAHITI): https://dahiti.dgfi.tum.de/en/products/river-tides/map/. Selected rivers are presented in Fig. 1.

We validated our tidal amplitude estimates from SWOT data using records from 622 globally distributed, in situ tide and river gauge stations. Median and mean amplitude differences for the 622 sites were only 5.53 cm and 9.23 cm for $M_2$ and 3.23 cm and 5.75 cm for $O_1$. In Fig. 2 we present a scatter plot of both SWOT and tide gauge estimates for both constituents, as well as the median error as a function of distance to river mouth. Additionally, we present the respective tide gauge amplitude errors in Fig. 3 for $M_2$. The error for $O_1$ is in Extended Data Fig. 1. We observe errors ranging between 0 cm and 30 cm across the globe, with the largest errors being observed at higher latitudes, where the effects of sea ice may influence the estimates. Amplitude errors

of both constituents consistently increase the further upstream the observations are made (Fig. 2), which is expected due to the decreasing signal-to-noise ratio as tidal amplitudes diminish[19].

The expectation, particularly with the current length of the SWOT time series, is that the $M_2$ tidal component can be estimated with better relative accuracy due to its large amplitude. For context, based on in situ observations, $M_2$ is on average 5.5 times larger than the $O_1$ tide. Although retrieval of the $O_1$ signal is possible from an aliasing perspective, its estimates are potentially more influenced by noise contributions from other (non-tidal) processes. Indeed, the $M_2$ tidal component has a median error of 15% whereas $O_1$ has a median error of 58%. Compared with the apparent 68th percentile error of individual SWOT water surface elevations (WSEs) at the reach scale (10 cm to 18 cm)[32], tidal amplitude estimates are highly accurate, especially considering the potential for uncertainties due to aliasing, which will only diminish as the SWOT record grows with time. These results confirm the possibility of deriving tides from RiverSP.

SWOT has orbited in two phases: (1) calibration and validation (Cal/Val) orbits and (2) science orbits. Some RiverSP nodes contain data from both types of orbit by the satellite. The Cal/Val phase of the satellite allowed the retrieval of tidal constituents using very few observations based on tidal aliasing with relatively high levels of accuracy[30].

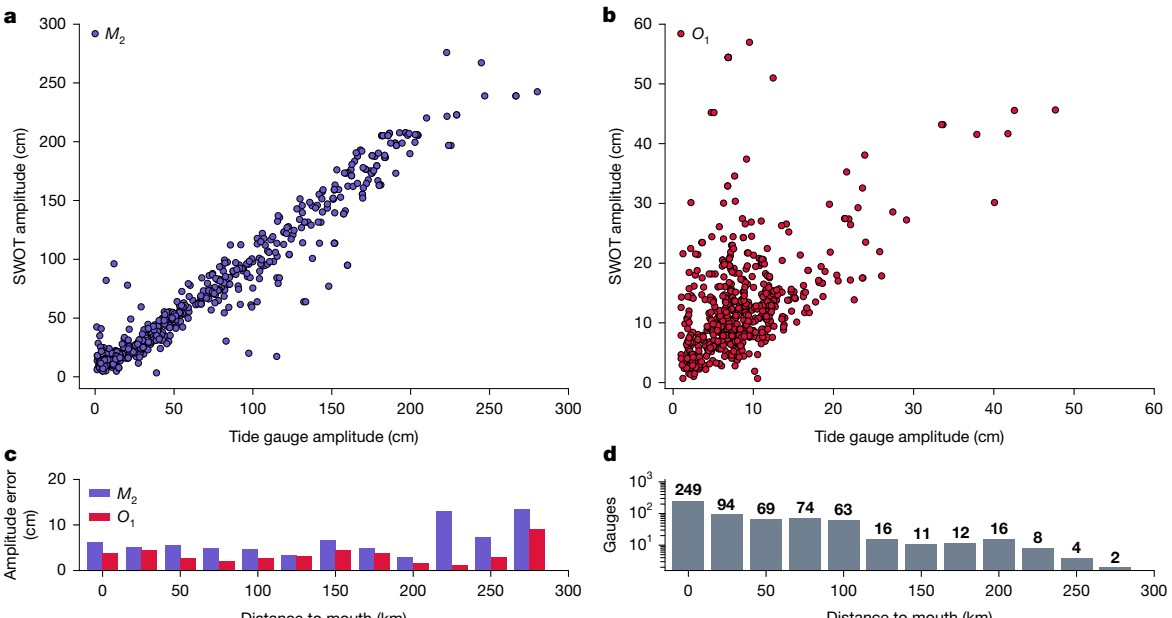

**Fig. 2 | Error statistics of SWOT tidal estimates compared with tide gauge observations. a,b**, Scatter of $M_2$ (**a**) and $O_1$ (**b**) amplitude estimates from tide gauges (*x*-axis) and SWOT observations (*y*-axis). **c**, The median error as a function of distance to the river mouth. **d**, The number of in situ observations used in the distance to mouth bins.

The median amplitude errors in reaches that contain both Cal/Val and science orbit observations are 5.74 cm and 4.28 cm for $M_2$ and $O_1$, respectively, whereas the errors are 5.54 cm and 3.29 cm when using only the science orbits. These differences between the median amplitude errors indicate that, although the combined Cal/Val and science orbits provide more observations, the results derived from RiverSP data from the science orbits alone are comparable with those combined with the Cal/Val orbits. An important caveat is that the Cal/Val phase does not provide a global perspective, as only 73 river tide reaches with in situ tide gauge measurements were covered, compared with the full dataset with 622 tide gauges.

The fidelity and unprecedented resolution of river tides from SWOT are evident in the dominant $M_2$ amplitude along three well-gauged rivers: the Gironde, Seine and Elbe Rivers (Fig. 1). The three selected rivers are sixth-order rivers[33,34] with a mean annual discharge around 500 m³ s⁻¹ (refs. 35,36), and they have slopes between

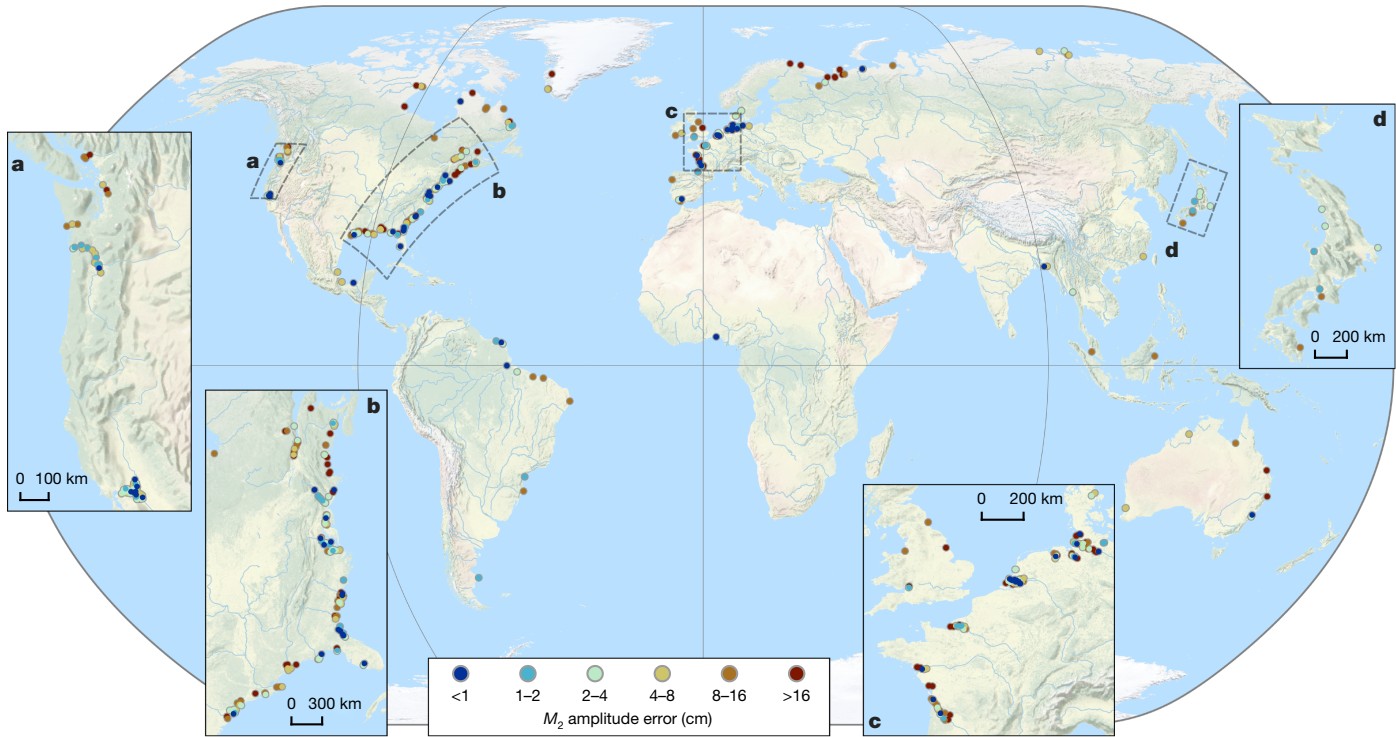

**Fig. 3 | The amplitude error of the $M_2$ tide mapped with respect to in situ observations from TICON-4, with several zoom-ins to different regions. a–d**, Regional zooms are provided into the west coast (**a**) and east coast (**b**) of the USA, the northern coastline of Europe (**c**) and the coast of Japan (**d**). The error for $O_1$ can be found in Extended Data Fig. 1. Base map from Natural Earth.

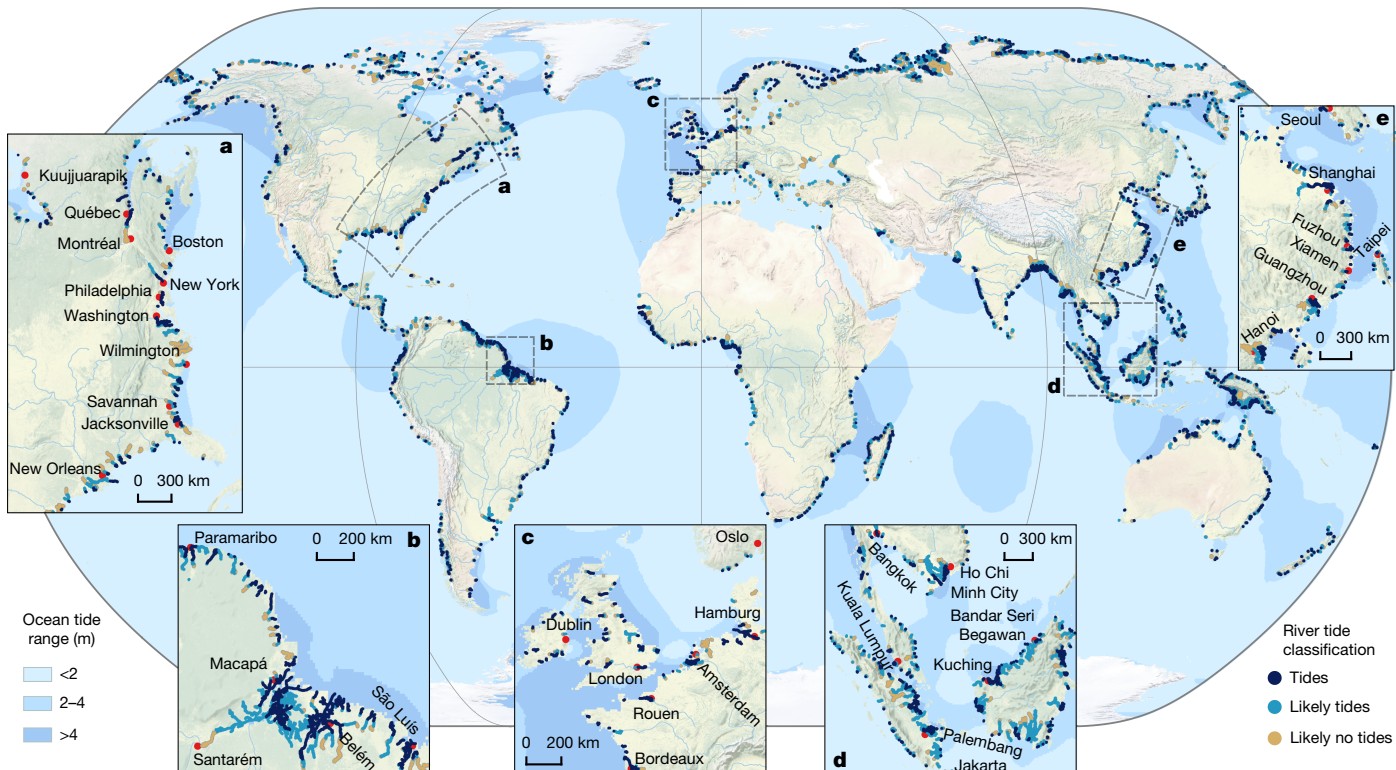

**Fig. 4 | Global river tide classification atlas with detailed inset maps of select coastal areas. a–e**, Rivers were divided into four classes (tidal, likely tidal, likely not tidal and not tidal), as described in Methods. Only the first three classes are plotted. An interactive map version is available at https://dahiti. dgfi.tum.de/en/products/river-tides/map/. Open-ocean tidal ranges were extracted from the EOT20 empirical tide model[6]. Red dots mark principal cities. Regional zooms are presented in the east coast of the USA (**a**), the northeast Brazilian coast (**b**), the northern coastline of Europe (**c**), part of south east Asia (**d**), and part of eastern China (**e**). Base map from Natural Earth.

10 mm km$^{-1}$ and 60 mm km$^{-1}$ over the studied sections[37]. The $M_2$ signal clearly persists over long distances in all three rivers. For example, the Elbe River has amplitudes greater than 50 cm at a distance more than 100 km upstream of the river mouth (Fig. 1c). Interestingly, the tidal amplitude in the Elbe River increases upstream of the river mouth near Hamburg in both the SWOT estimates and the tide gauge data. This amplification is well documented and is a result of the convergence of river branches upstream of the river mouth[38]. Another example of complex tidal variations is evident in the Seine River (Fig. 1b). The tidal amplitude slowly decreases from the mouth, increases again in the meanders further upstream and then declines sharply near a dam, which coincides with the in situ observations. These tidal amplitudes demonstrate the complexity of the propagation of tidal waves in channelized water bodies and the need for observational approaches that can be coupled with tide modelling strategies.

A direct comparison between the reconstructed water levels from SWOT and in situ observations from these three rivers further illustrates the fidelity of the SWOT analysis (Extended Data Fig. 2). We selected four gauges and associated SWOT time series where we have a combination of Cal/Val and science orbit data as well as profiles classified as 'tidal', 'likely tidal' and 'not tidal' (these classifications are described in Methods). Using the appropriate tidal amplitudes and phases, we derived estimates of tidal height and compared these against the tide gauge and raw RiverSP time series. Two of the three profiles, identified as tidal, match reasonably well for all three time series, indicating a clear dominance of the ocean tides in these regions. In the likely tidal region, the tidal signal is still observable, but this time series is more influenced by non-tidal effects. In the not tidal series, no substantial tidal variability is observed or calculated from our dataset. This comparison exemplifies the importance of the new SWOT dataset for estimating tidal heights, which can be used in combination with ground-based data to help understand the role of tides on local river processes or to correct time series to study non-tidal processes.

The analysis of the Seine and Elbe Rivers also illustrates the effects of dams (white squares in Fig. 1). Tidal amplitudes before the dams exceed 50 cm in both regions, whereas the tidal amplitudes upstream of the dams are below our detection limit of 10 cm. The Garonne River within the Gironde Estuary (Fig. 1a) is unique as it does not contain an obstacle affecting the tidal extent. The tidal amplitude naturally dissipates to below 10 cm at approximately 230 km from the mouth upstream of Langon due to friction with the riverbed[19].

## Global classification of tidal rivers

Worldwide, we studied 51,627 river branches and identified more than 165,000 km of river extent that contain the pulse of the tide (Fig. 4). The rivers are classified as tidal, likely tidal, likely not tidal and not tidal (Methods). Tides are shown to extend for hundreds of kilometres in many of the world's rivers and are particularly evident in the Hudson River (Fig. 4a), Amazon River (Fig. 4b) and Yangtze River (Fig. 4e). The Amazon River is an extremely complex river system and is well known to be influenced by ocean tides[39]. In situ measurements within the Amazon show tidal amplitudes exceeding 10 cm over large sections[40], with the $M_2$ amplitude dropping to 2.3 cm near the Óbidos region, about 892 km upstream of the river mouth. Our automatic approach determined that the tide within the Amazon stops between the Santarém and Prainha regions. This correlates with the tide gauge results[40], which show that the tidal amplitude drops below 10 cm in the same region. This result demonstrates the suitability of this approach and the RiverSP data in determining tidal extents, even in complicated river systems with extensive floodplains, dense vegetation and branching channel networks.

Within the European region (Fig. 4c), tidal rivers are more pervasive and longer in the north-western part of the continent than in the

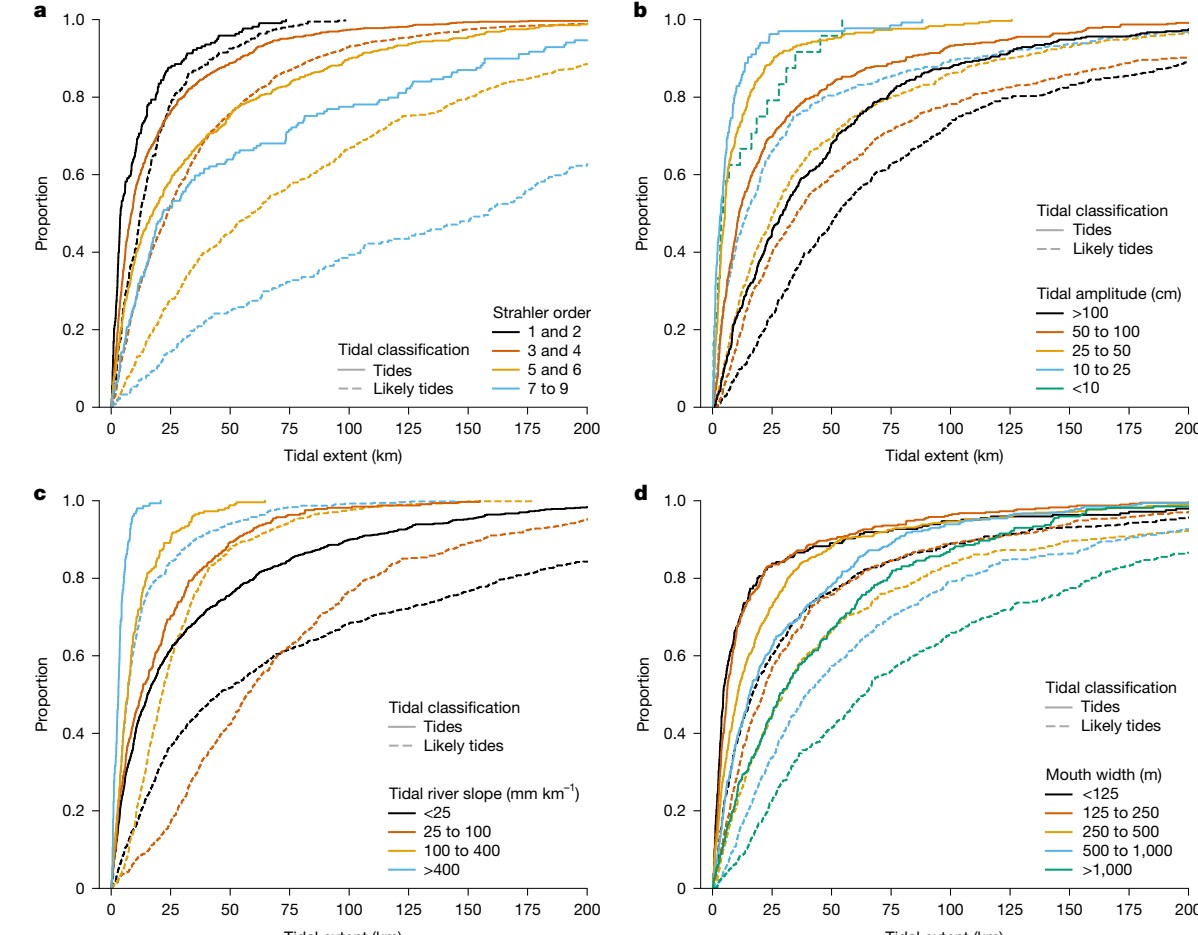

**Fig. 5 | Statistical distribution of tidal extent grouped by river and tide characteristics. a**–**d**, Cumulative density functions of tidal extent per estuary grouped by tidal likelihood and Strahler stream size (**a**), tidal amplitude at the river mouth (**b**), average tidal river slope (**c**) and river mouth width (**d**) obtained from SWORD. Tidal extent was determined as the distance along the river from its mouth to the farthest head of tides in any of its tributaries.

southern and eastern parts, consistent with the larger tidal ranges in the Irish Sea, English Channel, North Sea and the Bay of Biscay compared with the Baltic Sea, Mediterranean and Skagerrak Channel. Notably, the estimated tidal likelihood, which is based on a set of threshold conditions, is reduced in Dutch rivers compared with French, British and German rivers due to the considerable effort by the Dutch government to implement flood protection methods, like barriers and dykes, along the coastline to reduce the impact of sea-level rise[41]. The sensor capability of SWOT limits observations to rivers with widths of at least 30–90 m. This results in the undersampling of smaller streams in our analysis. For comparison, Tagestad et al.[7] used static river elevation profiles to infer that 106,000 km of tidal rivers may exist in the contiguous USA alone, over half of which are small streams (first and second order). Although our estimated 165,000 km of tidal and likely tidal rivers is conservative, the analysis provides a record of tidal dynamics in thousands of ungauged coastal rivers worldwide.

For the 3,172 tidal and likely tidal river systems with sufficient data in the SWOT River Database (SWORD), we examined the relations between the tidal extent and the hydraulic and morphologic characteristics of the river system. Tidal extent was estimated as the distance from the tidal river mouth to the farthest head of tides in any of its upstream tributaries. We also estimated the amplitude of ocean tides at the mouth as defined by RiverSP, the river width at the mouth, the average slope of the mean WSE of the tidal river from river mouth to head of tides, and the Strahler stream order at the mouth (a measure of the size of the river network, where larger numbers indicate exponentially more upstream

confluences) based on the MERIT-Basins dataset[34] (Fig. 5). Overall, the largest rivers, like the Mississippi, Amazon and Nile (Strahler stream orders over 7), have a median likely tidal extent of 161 km, whereas the smallest rivers (Strahler orders under 3) have a median likely tidal extent of only 13 km (Fig. 5a). Within a given Strahler classification, tidal extents vary broadly due to the unique characteristics of each river. The tidal amplitude at the river mouth and slope both exert important controls, as bigger ocean tides propagate farther along low-gradient rivers[42,43] (Fig. 5b,c). For example, the median likely tidal extent in rivers with a slope of over 400 mm km[-1] is only 10 km, whereas the median likely extent in rivers with a slope of under 25 mm km[-1] is 44 km (Fig. 5c). The interaction of the tidal wave with steep gradients causes a faster dissipation of the tidal energy. Similarly, narrower channels also dissipate tidal energy over shorter distances. The median likely tidal extent is 67 km for river mouths wider than 1,000 m but only 17 km for river mouth widths under 125 m (Fig. 5d).

Across the entire global catalogue of tidal rivers, the extent of tidal influence scales roughly with Strahler order and, to a lesser degree, slope and river mouth amplitude (Extended Data Fig. 3). It is somewhat surprising that these network-scale river characteristics account for much of the global variability in tidal extent across small to large rivers. Much of the remaining variation arises from local complexities in tidal wave propagation, which depend on changes in slope and river width and the partitioning of flow between branching channels[19]. Interestingly, within our global assessment, the investigated rivers exhibit complex tidal characteristics, like upstream amplification,

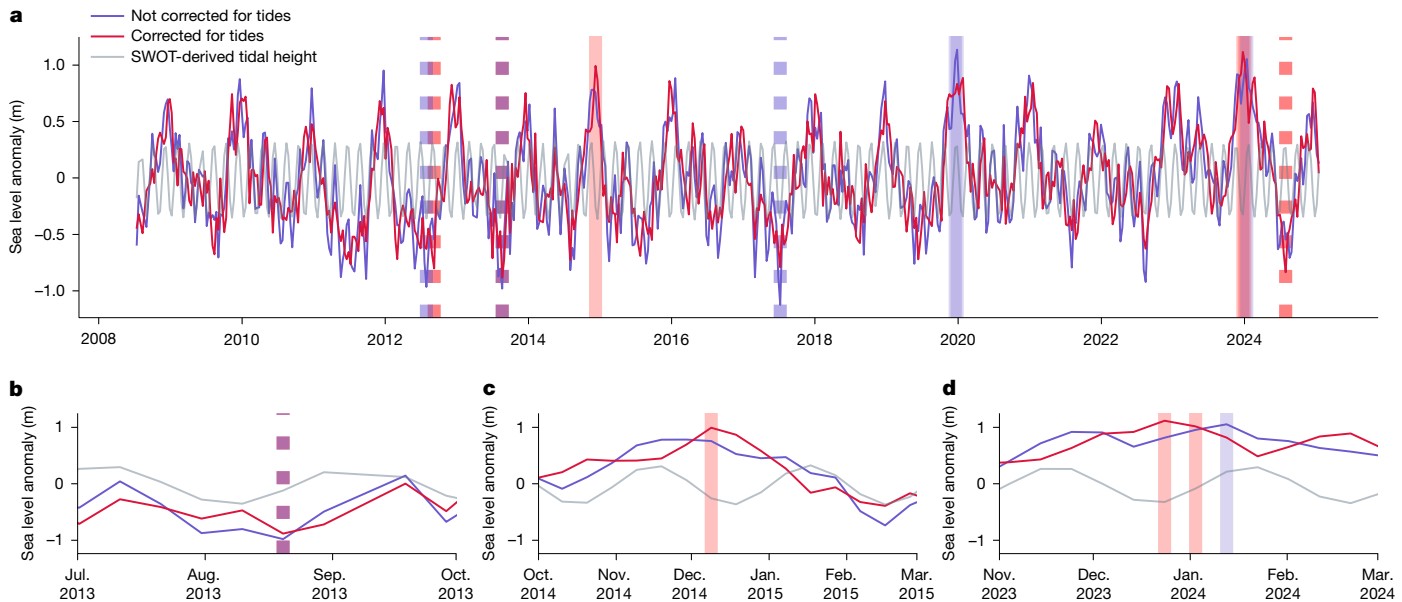

**Fig. 6 | Time series of satellite altimetry observations for an extreme value analysis of the Congo River. a–d**, A time series of satellite observations from DAHITI (purple lines). The tidal height is derived based on SWOT observations (grey lines), with this applied to the DAHITI time series to correct the $M_2$ and $O_1$ contributions (red). An extreme analysis was used to identify extreme low (dashed) and extreme high (solid) water levels based on uncorrected (purple) and corrected (red) time series. In places where both purple and red are present, the colour of the bars is indigo. Panel **a** presents the full time-series, while panels **b**–**d** are focused zooms into identified extreme water events.

which occurs in approximately 25% of tidal rivers, including the Elbe and the Gironde (Fig. 1). Despite these local complexities, the general relation between tidal extent and easily measured river characteristics, like Strahler order, reveals an underlying self-similarity among tidally influenced rivers. This result highlights the utility of observing patterns in tidal propagation in rivers globally based on SWOT wide-swath measurements.

River regulation and management also have a profound impact on tidal extent. Based on a comparison of our tidal river atlas and the database of dams and other river obstructions built into SWORD, 16% of the observed estuaries have an artificial or natural structure that limits the tidal extent (Extended Data Fig. 4). For comparison, approximately half of all rivers globally (both inland and coastal) have diminished connectivity due to dams and other structures[10].

Many of the known obstructions lie in the interior of river basins within the likely tidal or likely not tidal classes, where the tidal signal is less prominent. Tides in the smallest (first order) and largest (eighth to ninth order) rivers are relatively unimpeded, whereas tides in rivers of intermediate size are more typically impeded. There may be practical reasons why the largest rivers seem to have disproportionately fewer obstructions near the coast: structures that impede tides would also tend to restrict traffic along these important trade routes. Meanwhile, in small streams, impoundments may be less commonplace and also more difficult to detect where they do exist. For all obstructed tidal rivers, the mean amplitude immediately downstream of the obstruction is 54 cm. In some rivers, such as the Elbe River (Fig. 1c), the tidal amplitude immediately downstream exceeds 100 cm. These changes in tidal amplitude indicate abrupt human-driven changes in the hydrology and connectivity of coastal rivers, particularly those of moderate size.

## Implications for tidal rivers

The observation of tides has wide implications across the land–ocean–aquatic continuum and opens new doors in the study of coastal processes, including the analysis of extreme flows in sparsely gauged coastal rivers. As an example, we performed an analysis of extreme high and low flows on the Congo River near the west coast of Africa using a 15-year dataset of remotely sensed water levels from DAHITI

(see Methods for more details). One challenge of the nadir-altimetry-based DAHITI record is its relatively low sample frequency and inconsistent timing with respect to river tides, which obscure the assessment of flood severity and timing. To address this challenge, we corrected the coastal Congo water-level record for tides using amplitude and phase information derived from SWOT (Fig. 6) and then identified the three highest and lowest water-level events in the corrected and uncorrected time series for comparison. In the uncorrected data, three extremely low flow events and two extremely high flow events were sampled at low tide and high tide, respectively. A correction for the tides indicates that more extreme events occurred within the record (July 2024 and December 2014). Moreover, a well-documented flood in December 2023 caused by heavy rainfall upstream of the Congo River[44] was identified as an extreme event in both records, but the timing of the peak was obscured by tides. The corrected time series identifies the peak 2 weeks earlier (23 December 2023) than the uncorrected time series, which correlates better with the documented flooding in the region. SWOT, thus, opens new pathways for improving the observation of historical floods and droughts and for determining their recurrence statistics along tidally influenced rivers like the Congo.

By combining SWOT observations with river discharge or water quality information, SWOT also creates opportunities to monitor and predict saltwater intrusion in coastal rivers. Saltwater intrusion is increasing in frequency due to intense water withdrawals and drought in a changing climate[45]. Saltwater intrusion threatens the drinking water supplies for large cities, irrigated agriculture and industries that depend on fresh water. Tidal signals from SWOT are critical inputs for the development of new hydrodynamic models and artificial intelligence algorithms that can help predict and manage saltwater intrusion in sparsely instrumented rivers.

Compared to worldwide population statistics[46], we determine that approximately 10% of the global population, or 715 million people, live within 10 km of a tidal-influenced river. For comparison, Edmonds et al.[8] estimated that approximately 340 million people live in river deltas, and global studies indicate that approximately 1 billion people live within 10 km of a coast[47]. Globally, 110,410 km² of agricultural cropland, or 2.5% of the global total in the Copernicus landcover data[48], lies within

3 km of a tidal-influenced river and is probably affected by ocean tides, which highlights the potential implications for cropland flooding, saltwater intrusion and landscape change as sea levels rise. Given the threats to freshwater resources in these valuable tidal waterways[45], SWOT provides a powerful dataset that can be used to understand water and food security for large portions of the global population.

## Conclusion

The ability to measure the tidal pulse of rivers from space marks a breakthrough in understanding the interplay between hydrological and oceanographic processes. The high-resolution wide-swath SWOT data enable the characterization of fine-scale spatial variability within the tides. Moreover, the lightweight vectorized RiverSP presents the opportunity for these data to be incorporated within global empirical models, such as the Empirical Ocean Tide model[6]. This opportunity paves the way for potential tidal corrections consistent with ocean predictions to be constructed for rivers, thus addressing a long-running challenge for the tidal research community.

By providing a global atlas of tidal inundation in rivers, this study also opens new doors to cross-disciplinary investigations at the land–ocean interface. As an example, we have shown that tidal amplitudes from SWOT can be used to refine the analysis of historical floods and droughts on tidal rivers like the Congo. Tidal data can also be incorporated into artificial intelligence algorithms used to manage tidal freshwater resources[45] or to assess global carbon emissions from coastal rivers, which are at present poorly constrained because they require knowledge of tidal variations in river surface area[7,49]. Tidal amplitudes from SWOT can also serve as critical inputs for morphodynamic models of coastal wetland responses to rising sea levels[8,50]. As we continue to learn from these data as the mission progresses, refinements will improve and extend the approaches implemented here. Notably, an extended time series of observations will allow for a greater number of tidal constituents to be determined and will allow complex tide–river interactions to be studied; the current time series is too short to resolve any dependence of tides on river discharges, for example.

In conclusion, the early data provided by SWOT provide an unprecedented opportunity to advance our understanding of the complex interactions within the land–ocean continuum. The demonstrated ability to resolve tidal dynamics throughout river profiles opens the door to understanding the effects of tides on complex hydrologic and ecologic processes and climate-related risks at the land–ocean interface, including floods and saltwater intrusion. The data and analytical approach can be integrated into model workflows to help identify at-risk coastal communities, both retrospectively and in real time, across the globe. As SWOT continues to orbit, potential future missions are launched and the community continues to improve measurement quality, these data will enable refinements to tidal estimates in coastal rivers around the world.

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

# Methods

## SWOT RiverSP data

The global SWOT River Tides dataset (v.1) is based on the SWOT Level 2 RiverSP v.C[51], which provides node-scale preprocessed WSE measurements from SWOT with ~200-m spacing along the SWORD v.16[52] river centre lines. The improved parameters (like dist_out or distance from river mouth) of SWORD v.17 are appended to the RiverSP data using the SWORD translation lookup table. For version 1 of our dataset, we used SWOT measurements taken between 3 March 2023 and 2 May 2025, including the 1-day repeat Cal/Val orbit, because the high frequency improves the tidal estimates, if available[30]. RiverSP data are distributed in granules split by the SWOT pass (half of an orbital revolution) and continent. If a granule was processed several times based on different versions being available, we prioritized files with the highest fidelity (reprocessing being higher than forward processing) and the latest minor release version.

The SWOT RiverSP data provide several variables that can be used to discard unreliable observations. Based on community recommendations, we filtered the data using the following criteria:
- Dark water fraction: dark_frac < 0.4
- Quality flag: node_q ≤ 2
- Bitwise quality flag: node_q_b ≤ 2,097,152
- Crossover calibration flag: xover_cal_b ≤ 1
- Climatological ice flag: ice_clim_f = 0
- Minimum area: area_wse > 0

The most significant criterion for outliers is the climatological ice flag, which removed 80.9 million (40%) of the 195.3 million RiverSP observations. Another 20.2 million observations (10% of the data) were removed due to the dark water fraction exceeding 0.4, which typically occurs with a calm, mirror-like water surface. After outlier rejection, 72 million (37%) observations remained.

These filtered data provide an accurate time series of observations, which we used in a harmonic analysis. However, despite these filters, outliers may remain within the time series, which could have a large impact on tidal estimates. To account for this, we implemented an outlier detection algorithm for each node to find and remove observations from the time series based on a robust 3-sigma test using the median absolute difference. This outlier test rejected another 4.6 million observations or 6% of the filtered data. Once these quality steps had been completed, we gathered all the valid WSE observations within the masked coastal regions (see section 'HydroBASINS') into a node-scale time series. Because of the 120-km-wide swath observed by SWOT, the actual temporal resolution at a node covered by overlapping passes could be smaller than the nominal 1-day or 21-day repeat cycle of the respective orbit. As of May 2025, there have been 27 observations per node on average.

The 68th percentile differences between in situ gauge data and SWOT WSEs at the node scale range from 10 cm to 18 cm (refs. 32,53,54). Sources of error in the SWOT data include residuals from corrections for the wet troposphere, dry troposphere and ionosphere; residuals in roll error corrected using a calibration at ocean crossovers; errors associated with misclassification of land areas as water, which can result in the inclusion of higher land heights in node WSE calculations; the lack of radar backscatter at off-nadir incidence angles when the reflections from the water surface are too specular (a phenomenon often called dark water), which can reduce data coverage and accuracy; specular ringing from strong signals at nadir, which can affect height errors in areas even relatively far from nadir; topographic layover, which can cause major height errors in some cases; and a number of other potential error sources. There is a detailed discussion of SWOT error sources in the *SWOT User's Handbook*[55].

## HydroBASINS

The HydroBASINS[56] dataset at Pfafstetter level 6 was used to select RiverSP nodes within coastal basins that are possibly affected by tides. The Pfafstetter coding system delineates and codes nested river catchments[57]. We processed all basins that are flagged as the most downstream sink of the respective river basin (HYBAS_ID = MAIN_BAS) and not flagged as endorheic (ENDO = 0), which are rivers that do not flow into the ocean. Some upstream basins of large river systems, such as the Amazon, were added manually because a larger tidal extent was expected. Basins around the Caspian Sea were discarded from the analysis. The first six digits of the RiverSP node identifiers represent the Pfafstetter level 6 code of the enclosing basin, so that all nodes required for processing can be identified by this HydroBASINS subset.

## Tidal analysis

We used the unified tidal analysis and prediction functions (UTide)[58] implemented in Python[59] to conduct a harmonic analysis to estimate the node-scale $M_2$ and $O_1$ amplitudes from the RiverSP WSE time series. We selected only these constituents, as they account for regions that are dominated by either diurnal or semi-diurnal tidal components, which vary geographically, as demonstrated by in situ observation databases. Additionally, this selection also accounts for restrictions based on the aliasing period of tidal constituents, which, in the worst case when we ignore potential crossovers, would require observations spanning 66 days and 53 days to produce an estimate of $M_2$ or $O_1$, respectively, in the science phase (or 12 days and 13 days in the Cal/Val phase), as well as a span of 266 days (262 days) to separate the constituents from one another. For the $M_2$ tide, this is typically the largest constituent, and, thus, is less susceptible to noise, and it will generally play a larger role than the smaller tides. The amplitudes of both constituents were summed to obtain the maximum possible amplitude. The UTide confidence interval was calculated as $1.96\sigma$, and we calculated the combined confidence interval as $\sqrt{CI_{M_2}^2 + CI_{O_1}^2}$. In the validation and visualization of the tidal amplitudes, we used these error estimates to mask out uncertain estimates. As discussed below, these error estimates are a crucial component of the tidal classification.

Extensions to classical harmonic analysis have been proposed in previous studies[60] to account for the effects of time-varying external forcings on non-stationary stage and tidal components (termed NS_TIDE), which can influence tidal predictions in rivers. Preliminary comparisons for a dammed river and for an unobstructed river between NS_TIDE and UTide indicate good agreement of the time-averaged tidal amplitude and extent estimates. Relative amplitude differences of 3.4% and 6.1% for the Seine (dammed) and Garonne (unobstructed) rivers were found. However, as the SWOT time series lengthens and key products, like river discharge, continue to mature, extending the present analysis to explicitly resolve non-stationary tidal components will be a suitable follow-up. Such an extension will enable a more comprehensive assessment of how non-stationary processes modulate tidal amplitudes and extents in rivers at the global scale.

## Validation datasets

We used four different datasets to validate our estimates of tidal constituents. The TICON dataset is a global TIdal CONstant database[61] derived through a harmonic analysis of high-frequency sea-level measurements made by in situ tide gauge observations obtained from GESLA[62]. For this study, an update was made to TICON, termed TICON-4[63], based on the new release of GESLA-4. This dataset contains data from tide gauges along the coast as well as within several rivers. In this study, we used only tide gauge observations that were within 100 m of a SWOT node measurement to reduce the effect of any small-scale variations, particularly along the coastal zone. Additionally, we used only observations that were at least 1 km upstream of the river mouth. In addition to TICON, river gauge datasets were obtained from the United States Geological Survey (USGS), ArcTiCA[64] and the HydroPortail (HPO). For the HPO dataset, stations were manually selected within the expected tidal extent of available rivers. USGS provides river gauge observations that have been predefined as being influenced by tides or not. We selected only gauges

that are defined as influenced by tides. Once the WSE time series had been collected for each gauge within these two datasets, the same tidal analysis as described for the SWOT data was conducted to extract the amplitudes of the $M_2$ and $O_1$ constituents. Finally, the ArcTiCA dataset provides tidal constituents at higher latitudes, above 60° N, estimated from a variety of sources, including TICON-4 and from the USGS. In this study, we used all available sources except those already in the other datasets, and we processed the dataset in the same way as we did the TICON-4 data. By combining these three data sources, we obtained 622 unique in situ observations for validation, 554 from the TICON-4 dataset, 25 from HPO, 3 from USGS and 40 from ArcTiCA. Note that several of the gauges available from USGS are available within TICON-4, and we removed duplicate tide gauges from our analysis.

## Node grouping
Although the tidal analysis was done on each node of RiverSP, the next step was to collect the nodes along each river branch and sort the data as a function of chainage (distance to the river mouth). This step required reliable topology information, which was obtained from the latest SWORD v.17 dataset. Despite using the latest versions, some attributes of dist_out, rch_id_up and obstr_type were manually corrected to ensure that the topologies were realistic. The upstream nodes were gathered for each node classified as a river outlet, based on the rch_id_up parameter, until we reached the head of a river or an obstruction mapped in SWORD. If there were several upstream branches, the downstream data were copied, so that the data were grouped from each river outlet to each possible headwater. This resulted in several cases where there could be overlaps within river groups, which increased the computational effort for the tidal classifications but also improved the classification within each available river branch.

## Tidal classification
Once the nodes had been grouped in branches, some were flagged as outliers, either because they contained insufficient observations (fewer than 20) or because they contained obvious outliers resulting in unrealistically larger tidal amplitudes, which could be caused by tidal flats, as SWOT often observes these as water[65]. For these nodes, the tidal amplitudes and their respective confidence intervals, as well as the WSE statistics (median and standard deviation), were interpolated along the river chainage. As mentioned previously, the derivation of tides was probably influenced by noise relating to interactions with hydrological processes and tidal aliasing. To mitigate these effects, a median filter was applied to the tidal and WSE statistics within a rolling window of approximately 4 km (20 nodes), which aimed to reduce the influence of remaining outliers and data gaps. This window size was selected based on several iterations, as overly small windows did not effectively account for outliers and overly large windows yielded data too smooth to capture the variations in tidal amplitudes.

Once preprocessed, the following four conditions were applied to obtain the tidal classifications in the studied rivers:

(1) Confidence interval: The summed tidal amplitude must be greater than its confidence interval for this condition to be set to 1. Once the summed tidal amplitude dropped below the threshold, all upstream nodes were set to 0.

(2) Signal-to-noise ratio: We took into account the standard deviations of the WSEs with respect to the summed tidal amplitude. This condition was set to 1 when the summed tidal amplitude was greater than 2/3 of the WSE standard deviations. Once the summed tidal amplitude dropped below this threshold, all the upstream nodes were set to 0.

(3) River topography: For each node $x$, we estimated the minimum WSE, $h_0(x)$, which represents the river topography (longitudinal profile), as follows:

$$h_0(x) = \bar{h}(x) - \sigma h(x) \times 1.96, \tag{1}$$

where $\bar{h}(x)$ is the median WSE and $\sigma h(x)$ is the WSE standard deviation. When the summed tidal amplitude was ten times greater than the defined river topography, the classification condition was set to 1. Once the summed tidal amplitude dropped below this threshold, all upstream nodes were set to 0. This definition accounted for rivers with gentle or steep gradients, where we expected the propagation extent of the tides to vary. Several experiments were conducted based on this threshold, with the selection of the final criteria being based on optimizing the tidal classifications within the relatively gentle slope but large tidal influence of the Amazon River.

(4) Amplitude threshold: The summed tidal amplitude had to be greater than 13 cm. This is a rather conservative threshold in an attempt to account for the accuracy of the WSE dataset itself, which early studies have indicated to be about 12 cm at the node scale[32,53,54,66]. Additionally, our validation presented above concluded a median amplitude error for $M_2$ of 12.94 cm, which also provides support for the selected 13-cm threshold. Our studies on using harmonic analysis to derive $M_2$ and $O_1$ in parts of rivers that have no tidal influence found amplitudes in the single and low double digits of centimetres. This was expected, as hydrological signals can be aliased into tidal amplitudes, particularly due to the current short time series of data available. For all nodes that exceeded the amplitude threshold, this condition was set to 1. Once the amplitude dropped below the 13-cm threshold, all upstream nodes were set to 0. Additionally, this condition was set to 0 once $h_0(x)$ was ten times larger than the maximum observed amplitude within the branch. This improved classification for cases when a residual amplitude of over 13 cm was observed far upstream.

The sum of these conditions for each node was divided by the number of conditions to derive the probability of tidal influence $p$. If there was a data gap of more than 20 nodes (approximately 4 km) and $p = 1$, $p$ was reduced by 0.1 upstream of the data gap. Following this, any obstacles within the river, such as dams or waterfalls, were accounted for by setting the upstream probability to zero. This was done as the harmonic analysis still produced amplitudes for $O_1$ and $M_2$, which were either only an error or an artefact based on aliasing with non-tidal processes. An example of the result of the tidal classification procedure is presented for the Dordogne River in Extended Data Fig. 5. Subsequently, the probabilities and filtered amplitudes were transferred back to the node scale. As most nodes are part of several branches, the mean probability $\bar{p}$ per node was used to classify the probability of tides as tidal ($\bar{p} = 1$), likely tidal ($1 > \bar{p} \geq 0.5$), likely not tidal ($0.5 > \bar{p} > 0$) and not tidal ($\bar{p} = 0$).

## Extreme value analysis of the tidal Congo River
We used our tidal classification data alongside a time series of nadir altimetry observations from DAHITI[67] to assess the impact of tides on a historical flood analysis in the Congo River on the west coast of Africa. For the tidal Congo River, there were 15 years of observational records from conventional nadir altimetry, but there was a lack of consistent in situ observations documenting extreme flooding events. Using the amplitude and phase lag information of the $M_2$ and $O_1$ constituents from SWOT, we predicted the tidal component of river level changes for the 15-year time series, which we then used to correct the altimetry time series (Fig. 6). Extreme low and high events were identified in both the corrected and uncorrected series as events that exceed three sigma from the median absolute difference of the water-level measurements, shown in Fig. 6 with vertical stripes or dashes.

## Data availability
The RiverSP data used to derive the tidal estimates are available at https://podaac.jpl.nasa.gov/dataset/SWOT_L2_HR_RiverSP_2.0. The SWORD v.17 dataset used in this analysis is available at Zenodo (https://doi.org/10.5281/zenodo.14727521)[68]. The tidal classification data are

available at Zenodo (https://doi.org/10.5281/zenodo.15223861)[69] and are visualized at https://dahiti.dgfi.tum.de/en/products/river-tides/map/. TICON-4 is available at https://doi.org/10.17882/109129. Basemap raster and vector data were retrieved from Esri and Natural Earth.

## Code availability

To conduct the tidal analysis, the publicly available UTide software was used, which is available at GitHub (https://github.com/wesleybowman/UTide)[58]. The code used to derive the classification as described in this paper and used to plot Figs. 1–6 is available at Zenodo (https://doi.org/10.5281/zenodo.15223341)[70]. Figures were plotted with QGIS and Matplotlib.

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

**Acknowledgements** This work was funded in part by the German Research Foundation (Deutsche Forschungsgemeinschaft) project TIDUS-2 (DE2174/12-2, Project No. 388296632). T.M.P. was funded by NASA (Grant No. 80NSSC25K7715). We extend our sincere thanks to the SWOT Team, NASA and CNES for their generosity in freely distributing the SWOT science data. We thank the USGS, HPO and GESLA-4 for providing the in situ data. We thank C. Chen at the NASA/Caltech Jet Propulsion Lab for input on SWOT performance characteristics.

**Author contributions** M.G.H.-D., D.S., C.S., T.M.P. and R.D.R. conceptualized this study. M.G.H.-D., D.S. and A.H.S. carried out the analysis and experimentation and created the figures in this paper. D.S. and M.G.H.-D. developed the appropriate software and produced the final data presented in this paper. M.G.H.-D., D.S., C.S., A.H.S., T.M.P. and R.D.R. wrote the first drafts, and P.M. contributed to a revised draft. M.G.H.-D., D.S., C.S., A.H.S., T.M.P., R.D.R., P.M., D.D. and F.S. reviewed and edited the paper.

**Funding** Open access funding provided by Technische Universität München.

**Competing interests** The authors declare no competing interests.

**Additional information**
**Correspondence and requests for materials** should be addressed to M. G. Hart-Davis.

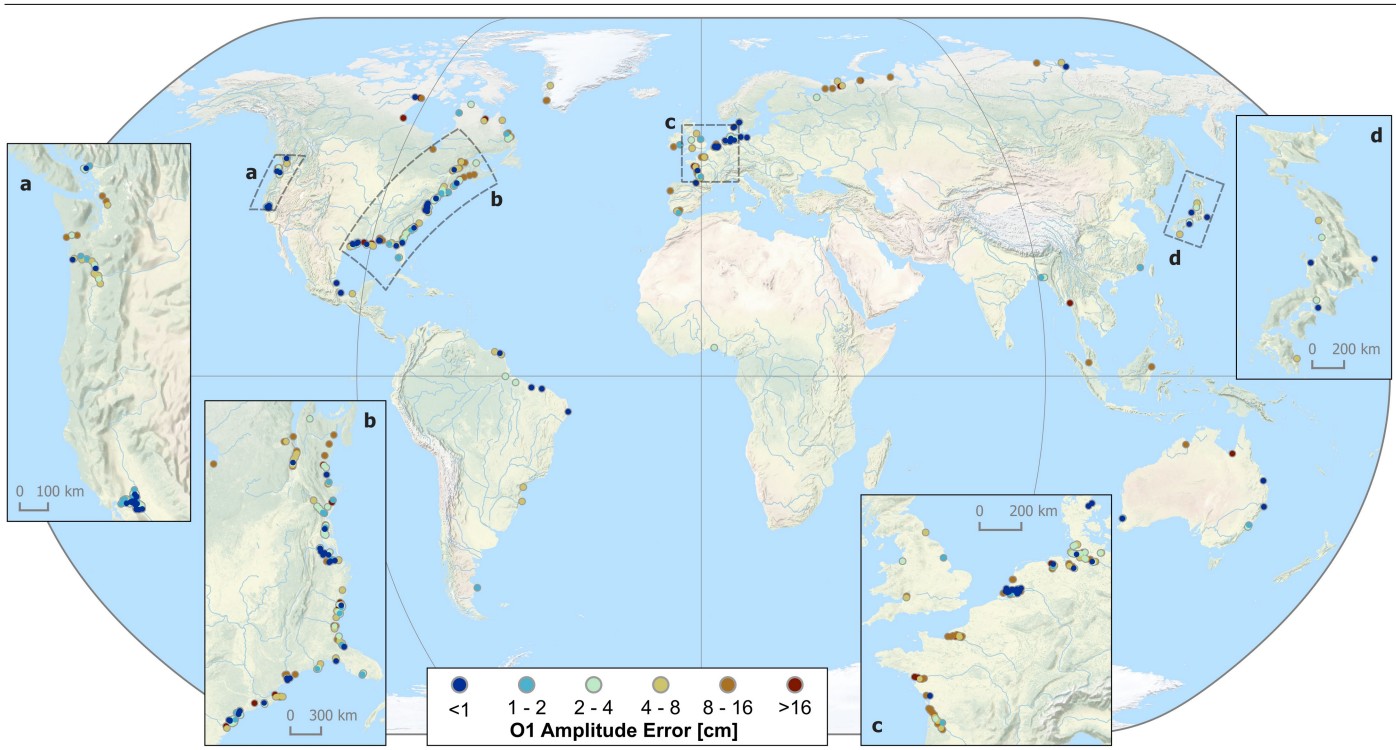

**Extended Data Fig. 1 | The amplitude error of the $O_1$ tide mapped with respect to in-situ observations from TICON-4, with several zoom-ins to different regions. Regional zooms are provided into (a) the west coast and** **(b) east coast of the USA, (c) the northern coastline of Europe and (d) the coast of Japan.** Base map made with Natural Earth.

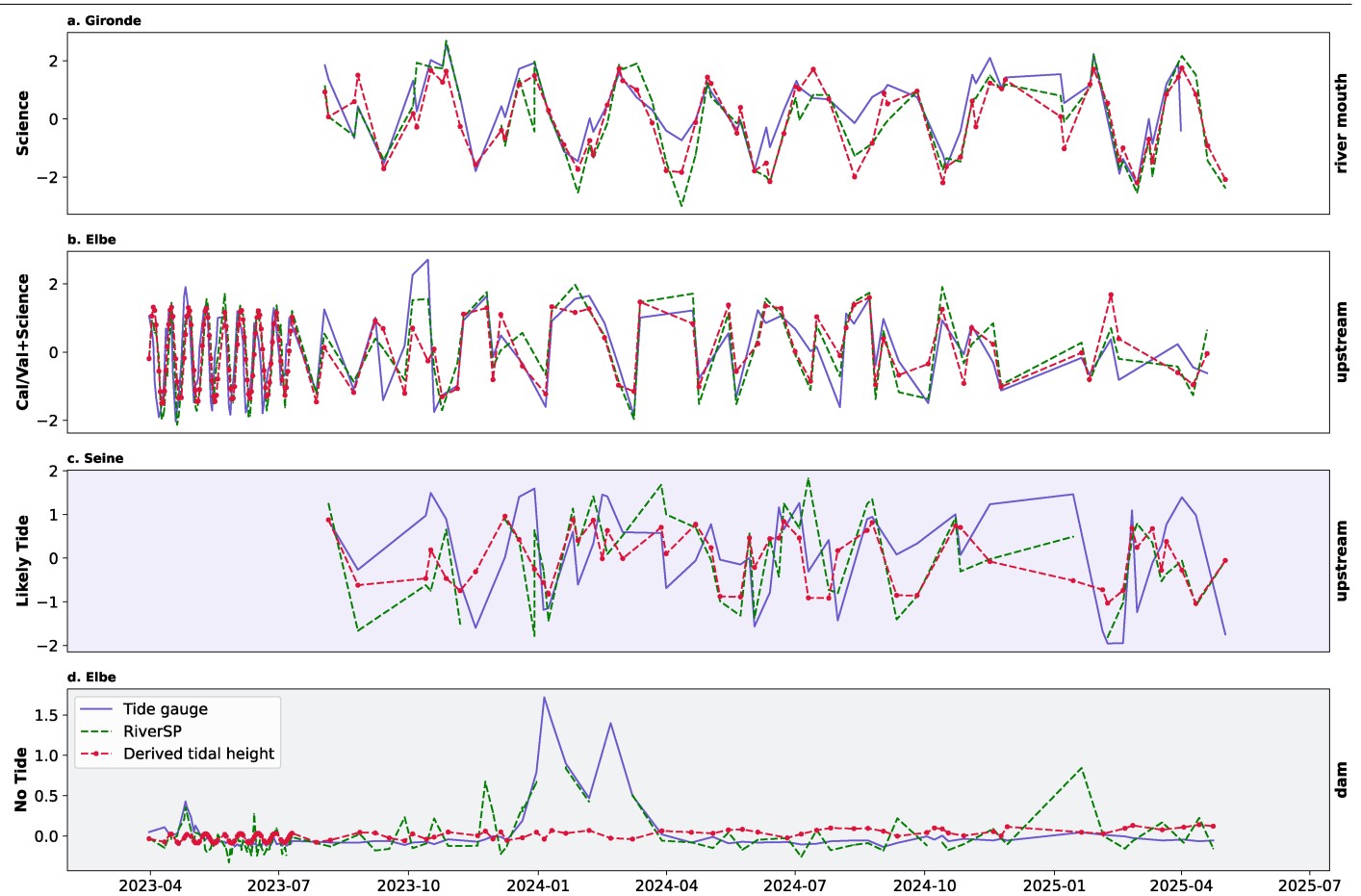

**Extended Data Fig. 2 | Time-series of SWOT and tide gauges in various river sections.** These time-series are along a river profile from river mouth upstream past a dam from the SWOT RiverSP product (green) and tide gauges (blue). The derived tidal height is presented for each location along the river profile. The selected rivers for this analysis are shown in Fig. 1.

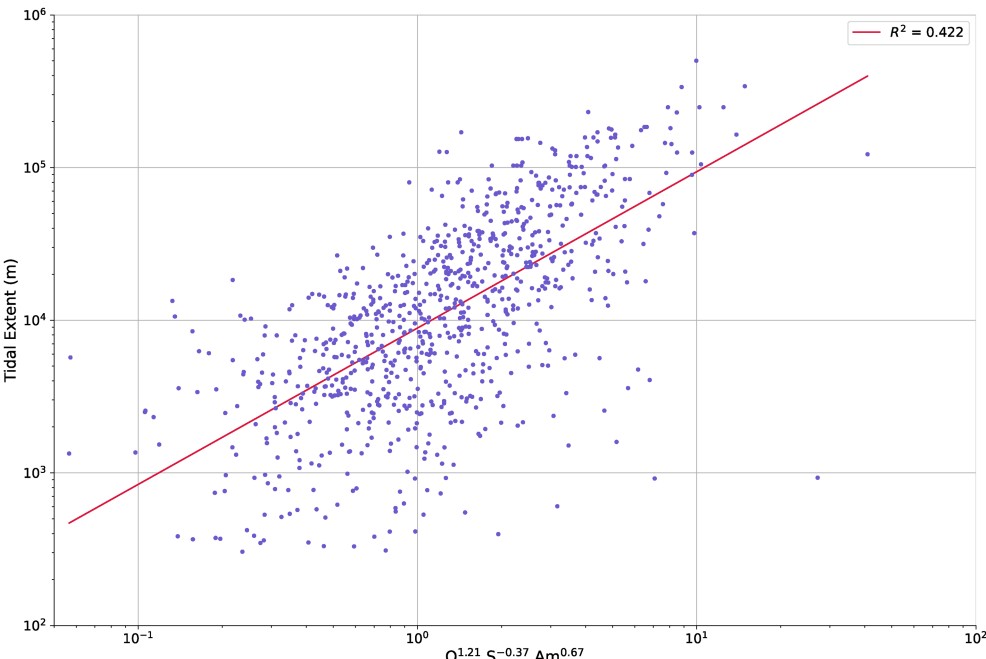

**Extended Data Fig. 3 | Empirical relationship between tidal extent and available river parameters.** These parameters, including Strahler order at the mouth (O), river slope (S), and river mouth summed amplitude at the mouth (Am), were derived from the global database. The scaling relationship on the x-axis was calculated from the individual correlation ($R^2$) between the logarithm of tidal extent and the logarithm of each of four individual variables: O, S, Am, and width at the mouth. The latter was not included in the final scaling relationship due to low correlation.

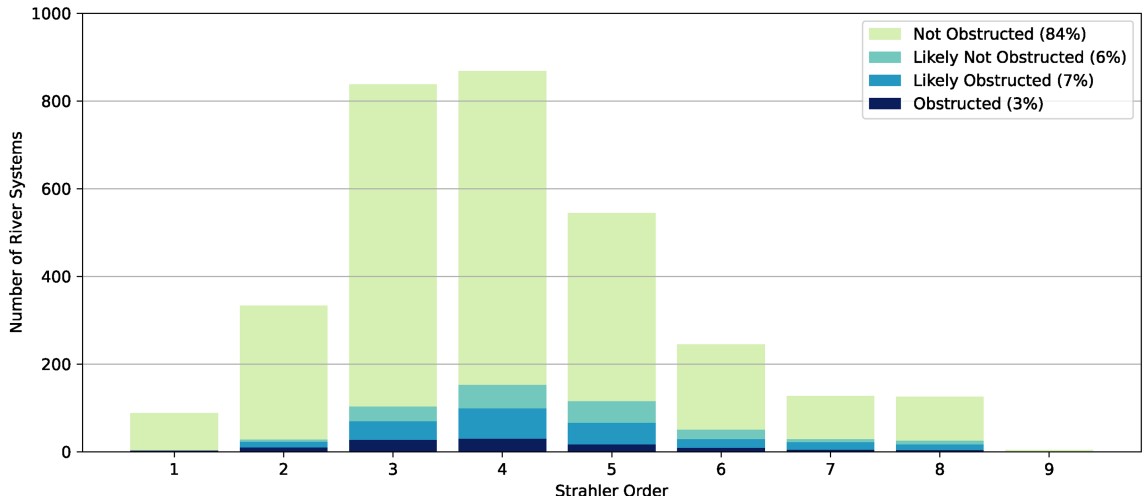

**Extended Data Fig. 4 | Histogram of the estuarine river systems grouped by Strahler order at the mouth, with the colors showing the probability of the tidal extent being obstructed by either manmade or natural features.** For Strahler order 9, there are two estuaries that are not obstructed.

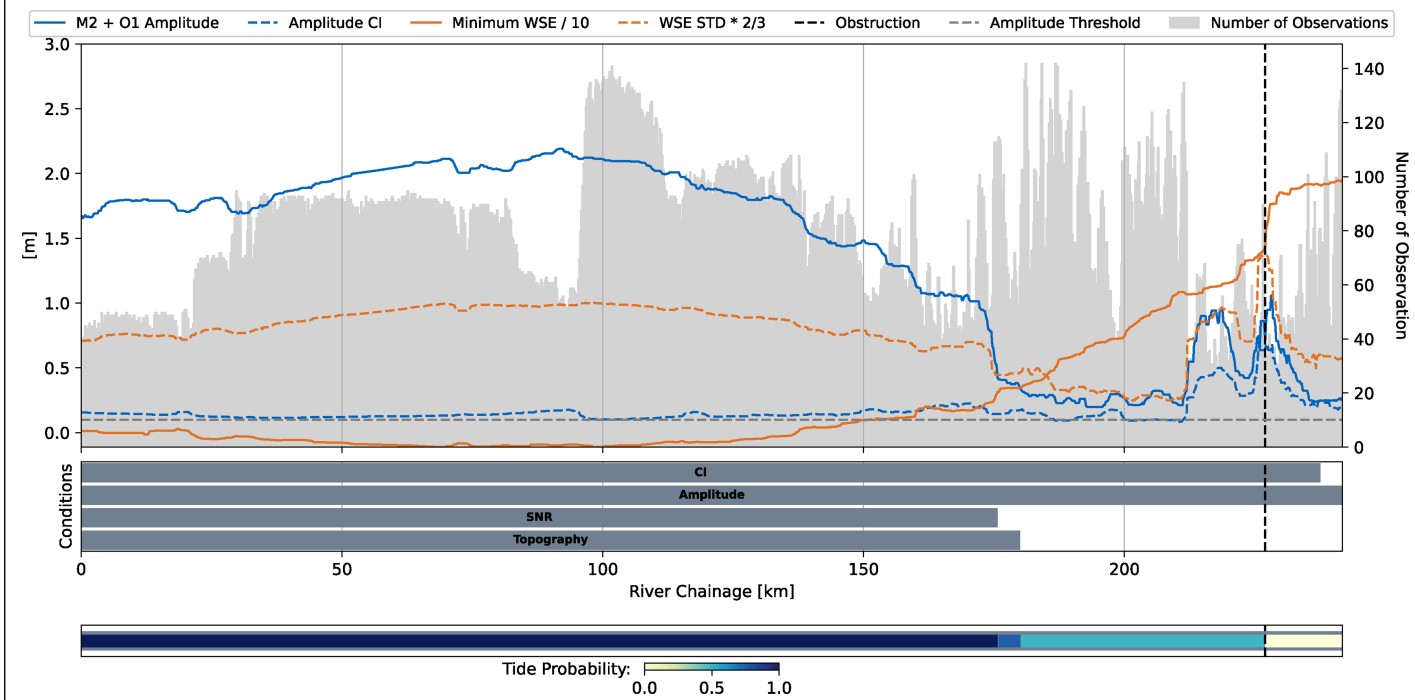

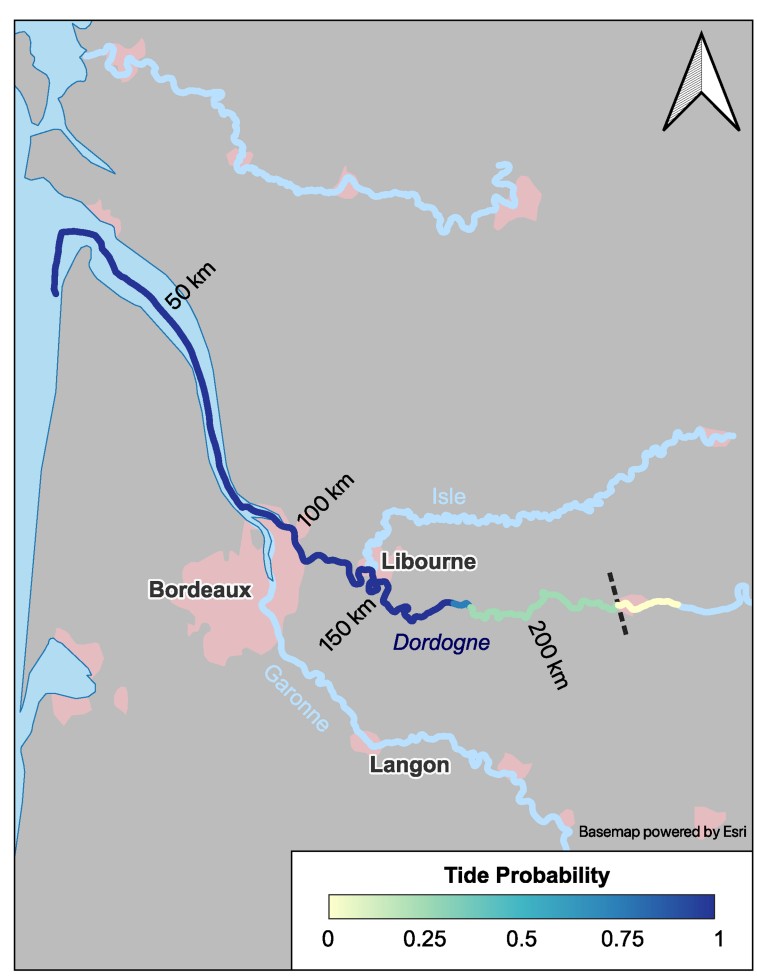

**Extended Data Fig. 5 | A graph of the $M_2 + O_1$ amplitude, confidence interval, minimum WSE, WSE standard deviation, and the number of observations within the Dordogne River and the tidal probability used to derive the tidal classification.** A map of the selected river branch and its tidal probability that was derived in this study. Base map made with Natural Earth.