## [Peer Review File · Nature]

Observing the tidal pulse of rivers from wide-swath satellite altimetry

Corresponding Author: Dr Michael Hart-Davis

Referee 3 was unable to accept a re-review. We therefore added co-reviewers 4 and 5 to assess the authors' replies and revisions to the now-missing referee.

Version 1:

Reviewer comments:

Referee #1

(Remarks to the Author)

This paper presents a first global atlas of observed tidal rivers, based on SWOT measurements. It links to a wonderful interactive web visualization of the identified rivers and associated metrics including tidal extent, amplitude, probability, and number of observations per node. Using the dataset, the authors examine the relationship between river characteristics and their tidal extent. This paper provides a world-first resource, and as such, an important and original contribution worthy of publication. Mapping the tidal characteristics of rivers globally, including regions remote or inaccessible, provides a rich resource for scientific exploration and discovery, and we expect it to be of interest to a wide-ranging readership. The paper is clearly written and well-presented (with a caveat regarding figures 3 and 4, see below). However, before it can be published, the authors should present a more detailed examination of the error budget. Second, the paper should provide more scientific insight – either directly or in the discussion of how the dataset can be utilized. Finally, the visualization of the data is uneven and should be improved upon to be suitable for a Nature publication.

A final comment – the Zenodo links provided for the dataset and the code were broken. We tried to search for them manually on Zenodo, but it appears they are not available, so we are unable to review these parts of the paper. This needs to be corrected in a revised edition.

Major points:

1. Error budget:

- Whilst exploring the web visualization, we noticed certain regions with abnormally large tidal amplitude far inland (e.g. east of Portland, OR). This appears to be an error but is labelled as 'likely tide'. If these sections of large amplitudes are not physical, can you try further quality control to remove the erroneous nodes?

- UTide is a robust analysis tool, well-suited for this kind of global survey; however, did you investigate using the NS_TIDE package for rivers where there is adequate discharge data to compare to results for UTide? (Matte et al. 2013, <https://doi.org/10.1175/JTECH-D-12-00016.1>) NS_TIDE is "adapted to the study of nonstationary signals, in this case, river tides. It builds the nonstationary forcing directly into the tidal basis functions. It is implemented by modification of T_TIDE; however, certain concepts, particularly the meaning of a constituent and the Rayleigh criterion, are redefined to account for the smearing effects on the tidal spectral lines by nontidal energy." If UTide tidal amplitudes are similar to NS_TIDE for well-constrained rivers, it would bolster global results for rivers where there are insufficient or non-existent discharge records. If there is substantial disagreement, this could help indicate how much error is introduced by neglecting the interaction of tides and river discharge.

- Certain rivers are covered by the CalVal orbit and so have the benefit of daily repeat measurements. It would be interesting to quantify exactly how much more accurately the CalVal-covered rivers are mapped than the science orbit-only rivers. It would give a sense of error for the science orbit-only rivers.

- Currently, you validate the remote SWOT data against in-situ tide gauges via a simple spatial comparison (Figure 1). This is a nice figure, but we expected to see a more robust validation, given the thrust of this paper is the dataset resource itself. Can you include Extended Data figure(s) showing a scatter of all tide gauge amplitudes vs. SWOT estimates. Also, you should present the locations of the tide gauges along river, because, for example, if all tide gauges are within a few kilometers of the river mouth, that will bias the validation results compared to, say gauges located O(100km) upstream, where amplitudes are much smaller and SWOT estimates likely more error prone.

- A time series comparison between river gauge data, SWOT data, and your estimated tidal components could be helpful to demonstrate how much non-tidal variability exists and how well SWOT matches in situ data; e.g. one case for different tidal probability regimes.

- You referenced studies, but it would be good to provide a concise explanation of the RiverSP error budget in the Method – i.e. what errors contribute to the error magnitude of 10-13 cm.

2. Scientific Insight.

- In the conclusions, you discuss briefly some areas that will benefit greatly from this new global dataset. It would strengthen the paper if you can give more specific examples of how the dataset can be used, and more examples: e.g. Can you discuss the predictive value of the dataset? e.g. erosion and sedimentation, long-term changes from SL rise etc.?

- Some other thoughts:

- o Can you say anything about river geometry (e.g. a metric of total curvature?) vs. tidal extent?

- o The paper discusses tidal amplification upriver – can we learn anything new about the characteristics of such rivers from the large number of tidal rivers now identified?

- o Given what you have learned about river characteristics and tidal extents, can you say anything about the thousands of rivers too small to be measured by SWOT but which may be tidally influenced?

- o Is there anything you can say about the characteristics of rivers that don't have tides?

3. Visualization of data

- Figure 3 is important because it presents initial insights from the new dataset. However, it could be a more impactful visualization of those insights – it took work to understand what information is being shown and disentangle the different variables and their relationships. Is a cumulative distribution function the best way to present the results? A scatterplot would directly show the relationships between river extent and each of the variables. If you have strong objections to a different type of graph, improving this figure would help – use better contrasting colors for the different categories, and don't cumulatively bin the data in addition to the cumulative y-axis (use discrete bins for the variables in panels b,c,d, as you did in panel a).

- Figure 4 does not appear to add anything new (the same histogram could be plotted from the prior datasets, since the Strahler order and obstructions are not new to this study) and could be moved to Extended Data. What is new is that you can measure the tidal amplitude change due to obstructions. Perhaps there's a better way to visualize that? Unless all obstructions reduce the tidal amplitude to zero and don't impact downstream variability.

- Considering multivariate data visualization, there are other types of graphs that could be employed for Figures 3 and 4 that may prove more fruitful, from scatterplots, box and whisker plots, parallel coordinates plots, radar plots, etc.

Minor points (general):

- Explain all specific terms when you first mention them (SWOT RiverSP product, river slope, Strahler order, Pfafstetter Level 6, etc.)

- A suggestion for the introductory paragraph following the abstract: Nature's wide readership might appreciate a short visceral description of what happens as a tide floods and ebbs in the river – describing the process as it unfolds and including in the discussion how the flooding water (for example) impacts water quality, or how the ebb impacts erosion etc. (i.e. make the discussion of tidal river importance more cinematic and compelling)

Minor points (by line number):

- Line 16: how do tides connect rivers with their surrounding floodplains? Can you be more direct, do you mean something like: "tides impact flooding of rivers and thus influence their floodplain extent"?

- Line 36: expand to 'launched in December 2022'

- Line 49: in this section you should add the time period over which the study was conducted (can move or repeat from Methods section)

- Line 76: I would rephrase to something like: 'Worldwide, we identify [count] tidal rivers and more than 150,000 km of river extent that contain the pulse of the tide'

- Lines 79-83: Can you explain the discrepancy between in situ measurements showing a tidal amplitude of 2.3 cm near Obidios, and the fact that SWOT measurements also show tidal amplitudes (>10cm presumably) near there?

- Line 92: Any surprises from un-gauged rivers measured for the first time by SWOT here?

- Lines 95-96: Was this including classification 'likely tides', or only 'tides'?

- Line 97: How did you estimate the amplitude of ocean tides at the mouth of the rivers? Was this EOT20 (Hart-Davis et al. 2021)?

- Line 100: This is the first mention of 'likely' classification. I think it would be helpful for the reader who doesn't want to get into the details of the Methods at the back of the paper to give a quick explanation of the categories tidal/likely tidal/likely non-tidal so that this paragraph makes sense on its own. Probably include it earlier at the beginning of this section or the previous section.

- Line 167: give context to term 'highest counter'

- Line 214: How do tidal flats result in larger tidal amplitudes? My intuition tells me that tidal flats provide a "floor" on the approximate sinusoid of a tidal time series, which would bring the trough up and reduce the total amplitude. A one-sentence explanation or citation could clear this up.

- Line 238: give reference here for the 'early study'.
- Line 241: Can you provide the frequencies that M2 and O1 are aliased to, in both the cal/val and the science orbit periods (whilst stressing, as you have elsewhere, that it's more complicated for each river node due to overlapping swaths).

Figures:

- Figure 1:
 - o In the legend the SWOT node is a black circle, but that is confusing since in the figure it is an unbroken line of color.
 - o Why have you not plotted M2+O1 amplitude here as you do globally?
 - o Can you plot the phase from both datasets? If so, it would be interesting to see, perhaps in Extended Data.
- Figure 2: Use a different color to separate 'tide' and 'likely tide' from the 'likely no tide' category.
- Figure S1: how do you arrive at the power law relationships plotted on the x axis? You need to fully explain.

(Remarks on code availability)

The links to code and data were not working, so they could not be reviewed.

Referee #2

(Remarks to the Author)

I co-reviewed this manuscript with one of the reviewers who provided the listed reports.

(Remarks on code availability)

Referee #3

(Remarks to the Author)

This manuscript is about investigating tides in coastal rivers on a global scale using satellite-based observations. The relevant scientific question authors like answering with this study is if the SWOT mission can be exploited to observe tides in 150,000 km of coastal rivers.

The estimation of the tidal field from space is not new in the open ocean and coastal zone. There is a coastal altimetry product coined X-TRACK Tidal Constant available at <https://www.aviso.altimetry.fr/en/data/products/auxiliary-products/coastal-tide-xtrack.html>. This product provides along-track tidal constant estimates for 73 constituents using long time nadir altimetry missions (TOPEX/Poseidon + Jason series) within almost all coastal seas in the world.

The novelty of this manuscript is the usage of the SWOT altimetry mission and the extension to the downstream part of rivers. The authors only estimate the diurnal O1 and semi-diurnal M2 tidal constituents, as they state the current SWOT record is too short. They use an high level product called "RiverSP" that provides preprocessed water surface levels spaced around 200 m. The estimation of tides is automatized in order to create a global atlas of tides within rivers. Validation of SWOT-derived tides is made using ground truth at 663 sites. Three rivers (Gironde, Seine, and Elbe) are analyzed in detail. Some other big rivers (Hudson, Amazon, Yangtze) are also highlighted.

Judging solely the presented results, the manuscript would be considered a ground-breaking in the field. However, I have some doubts that SWOT is capable of providing such stated stunning results. Overall the authors do not convince me. I see two important weaknesses: 1) the analyses of SWOT altimeter data is substantially absent considering that there are many caveats for this mission over inland waters; 2) the validation of SWOT-derived estimates of tides is poorly addressed.

SWOT is essentially a SAR system looking down at very low incidence angles. It uses the interferometry principle to estimate surface level changes. The KaRIn instrument permits to acquire Single-Look-Complex images. Then a product called "PIXC" is derived that is the input for the "RiverSP" product used here. The algorithm to estimate surface level elevations is extremely complex. There is a static mask of prior water probability used during processing that informs where water might be expected. It is not perfect for two reasons: a) it cannot be perfect as water is dynamic; b) the transformation of ground coordinates to slant-plane coordinates is also uncertain and depends on the water height itself. Therefore, there is an impact in determining if a pixel is water or land or mixed.

The power of echoes is also important in the classification. The document titled "SWOT Science Data Products User Handbook (JPL D-109532, Revision A, March 2025) states that "water detection for HR data over land assumes that water is brighter than the surrounding land". An example of the Connecticut River is provided (page 169). If the water is not much more reflective than land, it may not be detected as water in the KaRIn ground processing. This would cause errors.

Other two important issues are: a) the frequent presence of smooth waters (known in SAR domain as dark waters) that reflect very little radar energy at the KaRIn off-nadir incidence angles; b) the brightness of side lobes (called specular ringing). There are other caveats (e.g., layover, phase unwrapping, etc.). Several quality flags are provided in the products and must to be carefully scrutinized.

Authors here just look at one flag if observations are "degraded" or "bad measurements". It is important to highlight that "flags" alert you about potential issues that might affect the accuracy of the data. They simply inform users to look carefully those data. But also data flagged "good" might be "bad". This is why a data analyses is necessary. SWOT is certainly an intriguing satellite mission, but the complex processing calls for a severe data screening and I don't see any analyses in the paper.

There are published papers that already provide some indications that data analyses and validation is important. For instance Do Amaral et al. (2024) state "only about half of the WSE and WSS measurements meet the error budget and both require in situ validation for reliability. Additionally, we identified phase unwrapping and layover as significant sources of error at river locations near the nadir gap."

F. Rodrigues do Amaral, T. Nguyen Trung, T. Pellarin and N. Gratiot, "Challenging SWOT: Early Assessment of Level 2 High-Rate River Products in an Urbanized, Low Elevation Coastal Zone," in IEEE Geoscience and Remote Sensing Letters, vol. 22, pp. 1-5, 2025, Art no. 1500605, doi: 10.1109/LGRS.2024.3501407.

With regards to validation, the authors state that "Mean amplitude differences for the 663 sites were only 12.9 cm and 8.0 cm for the M2 and O1, respectively." These values are of the same magnitude of the error of SWOT water surface elevations (10 to 13 cm). Statistics might not represent what happens at single sites. The performance is highly variable from site to site and from day to day. Validation is key step before exploiting a data set or scientific studies. Validation cannot be reduced to a single number.

I looked at interactive map of Figure 2. I looked at the Mediterranean Sea that exhibits a micro-tidal regime. M2 variability is available from Figure 1 at <https://os.copernicus.org/articles/20/1051/2024/#&gid=1&pid=1>

My eyes immediately noted the River crossing Rome (Italy). It shows M2+O1 amplitudes higher than 2 m that is clearly an artifact of the processing, as those values are not realistic in that area (see above figure).

A second example is the Ebro River (Spain). Ibañez et al. state that "The effect of tides is noticeable as far away as Tortosa, 45 km from the mouth". But the map indicates "likely no tide" at that place. Tidal amplitude at mouth is predicted around 8 cm. Mestres et al. show that the maximum astronomical tidal range in the area is about 25 cm, with an average value of 16 cm.

Ibañez, C., Pont, D., & Prat, N. (1997). Characterization of the Ebre and Rhone estuaries: A basis for defining and classifying salt-wedge estuaries. *Limnology and Oceanography*, 42(1), 89-101.

Mestres, M., Sierra, J. P., Sánchez-Arcilla, A., Del Río, J. G., Wolf, T., Rodríguez, A., & Ouillon, S. (2003). Modelling of the Ebro River plume. Validation with field observations. *Scientia Marina*, 67(4), 379-391.

A third example is the Po River (Italy). SWOT data predict "likely tides" with "50% probability" at Polesella (74 km from the mouth) with M2+O1 amplitudes around 60 cm. Moreover, the atlas predicts a tidal amplification along the river. But Nones et al. shows in Figure 2 tidal influence along the Po River and there are no tides at Polesella and no amplification along the river.

Nones, M., Maselli, V., & Varrani, A. (2020). Numerical Modeling of the Hydro-Morphodynamics of a Distributary Channel of the Po River Delta (Italy) during the Spring 2009 Flood Event. *Geosciences*, 10(6), 209. <https://doi.org/10.3390/geosciences10060209>.

The previous examples show questionable results probably related to artifacts of the processing (e.g., dark waters, geolocations errors, layover, ringing, etc.). However, a deeper examination of SWOT data is absent. The authors state "each river has unique morphologic and hydrologic conditions". This means that the radar response is extreme variable. An in depth analyses of PIXC data, backscatter for each river, flags, radar scenario, etc. is necessary.

Also, the paper has major focus on a data set and some statistics. The reader would expect to see some new scientific insights from tidal fields in the rivers (e.g., tidal bore phenomena; tidal field propagation velocity; relationship with drought/flooding extreme events; salinity intrusions, etc.). But not a speculative statement like "Interestingly, within our global assessment, more than half of the investigated rivers (54.36%) exhibit e.g. upstream amplification similar to the Elbe". If tides amplify as they travel upstream river it depends on local conditions. For instance I have doubt this happens e.g. if the river mouth is wide and open to the sea without much constriction or e.g. if there is high discharge. Anyway, the above statement must to be proved.

A global atlas of tides from space in coastal rivers is certainly of interest to a wide audience, however, the analyses need to be rigorous. The reader expects that 1) SWOT observations are scrutinized in depth before estimating tides; 2) the tidal estimates are explained keeping in mind the relevant physical processes occurring in the selected sites.

The tidal classification data is available at Zenodo (<https://doi.org/10.5281/zenodo.15223861>)

The code to derive the classification as described within this manuscript is available at Zenodo (<https://doi.org/10.5281/zenodo.15223342>). However, both links do not work so I could not look at the data and code to check if paper's findings can be replicated.

Statistics should be more robust, e.g., validation is poorly reported (only a mean value and unknown how water gauge minus altimetry varies from site to site). Figure S1 has data points widely scattered and error of correlation is not reported.

(Remarks on code availability)

I was not able to access data and code (probably there is a mistake in the link).

Version 3:

Reviewer comments:

Referee #1

(Remarks to the Author)

Review of: "Observing the pulse of tidal rivers: A first global analysis from wide-swath satellite altimetry" by Hart-Davis et al.

Summary:

The authors have made significant changes to the manuscript, which have improved its quality and addressed most of our concerns regarding the dataset's error budget, paper's scientific insight, and visualization of the data. We recommend this paper for publication in Nature. Below we have included our remaining concerns, which we believe would strengthen the paper for posterity.

Major Points

Using NS_tide

We agree that a comprehensive examination of rivers using NS_TIDE is infeasible and best saved for a later investigation; however, it is beneficial to compare a small number of well-constrained rivers—even just one—to see if the difference between UTide and NS_TIDE is enough to cause concern with the amplitude and/or error estimates. This is what we meant in our original comment. We understand that the uniqueness of each river means that no global conclusions should be drawn from checking a few select North American or European rivers, but the best river tide analysis tool out there is NS_TIDE, so regionally validating the global analysis with it would be insightful. If you have a strong and compelling objection to this, you should at least include an explanation of why NS_TIDE is not practical in the paper, e.g. in Methods → Tidal Analysis, where UTide is introduced.

Reviewer 3's concerns on errors

We note that the website visualization still shows the river in Rome, Italy, with amplitudes > 3m, which apparently is unrealistic based on the reference provided by Reviewer 3.

Figures in main manuscript

Each figure of the main manuscript should be impactful, which we currently don't feel is the case. Figure 2 is good but has limited information—Figure S2, which provides more clarity on errors, should replace it; Figure 2 would be best placed within Figure 3 as a multi-panel plot, as it emphasizes the value of SWOT to move from very limited in situ observations to a truly global dataset. Finally, Figure 5 should be improved—it is not clear enough in its current format. A solution is to make it a multi-panel plot with zoom-in panels to better show the extreme high/low events. See minor points below for more comments on figures.

Minor Points

Title

The first line of your conclusion is "...to measure the tidal pulse of rivers from space...". Your title is "Observing the pulse of tidal rivers...". Our preference is to use the former in the title ("Observing the tidal pulse of rivers...") as it reads clearer for a general audience.

Figure 3

We disagree on the point of color palette. The green-blue of 'likely tide' and blue-green of 'likely no tide' is very hard to visually distinguish on the global map, and incorrectly implies the tidal rivers are longer than they likely are. Use yellow or orange (or another suitably different color) and then there will be 3 distinct colors for the 3 distinct categories.

Figure 4's line color choice:

Thank you for changing the bins, which we feel clarifies the results. It is indeed challenging to choose a good color scheme, but a "thermal" style colorbar like that used here is too difficult to distinguish between, especially for the 5-binned results in Figures 4b and 4d. A more distinct colormap such as "rainbow", "jet", or "turbo" will still possess an intuitive gradient but also better visualize the different lines.

Figure S2

Plot M2 and O1 on different panels. In its current form it's difficult to distinguish the O1 data, it looks like the scatter shows little relationship between SWOT and tide gauge, but that might be because the density of points are not visible.

Figure S3

Add y-axis values to show magnitudes.

Aliasing periods of tides in SWOT data:

Thank you for stating the aliased periods for both cal/val and science orbits. However, it is phrased confusingly: in Zaron (2024) <<https://doi.org/10.1029/2024EA003677>>, which only focuses on the cal/val orbit, it is more clearly stated that 12 (M2) and 13 (O1) are the "Tidal Alias Periods" in days. As it is phrased in your paper, there is an ambiguity between observational window and observation count (i.e., does the M2 need 66 snapshots from separate days from the science orbit to be observed, or just 4 flyovers that span a period longer than 66 days? I know that it's the latter, but it took a little bit to parse).

(Remarks on code availability)

Referee #2

(Remarks to the Author)

I co-reviewed this manuscript with one of the reviewers who provided the listed reports.

(Remarks on code availability)

The code linked above is not available; I am met with the message "Permission required - You do not have sufficient permissions to view this page."

However, the paper itself does link to a valid repository at <https://zenodo.org/records/15223342?preview=1&token=eyJhbGciOiJIUzUxMiJ9.eyJpZC6IjOiODcyNzBmLTczMmEiNDY1Yi04ZTBiLTlyZDc5NGEwNjg1OCIsImRhdGEiOiNlZCJyYjW5kb20iOiZyZjZlY2YzNiMThjZTlyNzZiZGY4NTA3NGVhODY5MmYyX3Q0OJ5-HAvz85qXBLtINyTQv1XcJHoUjw-tR241sJH2zfrK0Xe3xpxtWBzgvYASi50kR2Lq2jX9R_s6g>

I have not run the code, I have only verified that it is available. It appears to be complete.

Referee #3

(Remarks to the Author)

This paper is a resubmission of a paper that underwent to a first round of review. The authors attempt to showcase the abilities of SWOT for estimating tides in coastal rivers on a global scale (175,000 river kilometers). Overall there is no fundamentally improvement following my important recommendations.

During the first round of review I highlighted that "the analyses of SWOT altimeter data is substantially absent considering that there are many caveats for this mission over inland waters". The authors didn't follow my recommendation with data analyses as I expected to see in the revised paper. Then I told them that "the validation of SWOT-derived estimates of tides is poorly addressed". In response to this point, the authors added more tide gauges and corrected for some errors, however, without specifying the nature of those errors. The validation approach in the paper is based on quantity rather than quality. It is evident that the global validation cannot be homogenous at the land/sea boundary. Any site has its own characteristics in terms of water dynamics, morphology, placement of tide gauge, etc. All tide gauges are of opportunity (e.g., hourly measurements means no almost coincidence in time with satellite) and not placed for a validation of satellite-based observations (e.g., in protected areas, therefore, the two measuring systems not observing the same dynamics). The authors added some cal/val sites, but there is no site-specific analyses to assess the error budget. The global results (Figure 2, S1 and S2) blur local details, i.e., they hide local variations. The authors state that SWOT-derived water surface elevations have 68 percentile errors in the range 10-18 cm. This means 32% of values are larger than those ones. Then the estimates of the tidal amplitudes are stated much lower 3-5 cm. This as "median error" that means something averaged blurring the local variations.

Another important point I highlighted was the particular radar acquisition scenario in inland waters that is totally different from open ocean. SWOT is a SAR and all SAR images over land show that water is "dark", i.e., not reflective. This is why land is always more reflective than water. Going near nadir does not eliminate the issue, as the radar is always looking off. SWOT can only measure if water is more

reflective than land. Is this true for 175,000 river kilometers at any revisiting time? The authors did not prove that. Waters are frequently smooth over rivers. The authors here use a high level product. They apply some flags (i.e. something objectively stated), then further apply a customized editing. There is no understanding of what happens in the processing chain from radar measurements to extracted information (tide amplitude). Therefore, it is really difficult to comment the results. Only using errors per site the reader would have a more convincing error budget assessment (that would include not only the radar, but also orbits, corrections, etc.). The authors agree with the issues but then they state that the global analyses show accurate results. As previously stated, the global analyses blur the site differences. As the tidal data set is the output of the paper the analyses has to be rigorous and convincing rather than approximate.

Finally, the authors definitely do not convince me when they state "By adding additional data quality filtering, we have substantially reduced the mean amplitude differences to 11.5 cm for M2 and 6.4 cm for O1. Note, however, that we now choose to present the median values in the manuscript rather than the mean values, since a few significant outliers substantially affect the mean."

The median is just one number and ignores how the rest of the data are spread. There is no physical explanation behind numbers. The median value loses detail about how errors behave across sites or conditions. During the first round of review I highlighted some examples of unrealistic amplitudes. I hoped authors analyzed in depth to understand reasons for unrealistic values. Now authors state that there were "errors" without explaining why and that the new data flagging and outlier detections solve. Just a quick look at the new map showed me other inconsistencies (e.g., Oued El Kebir (Algeria) with M2 of around 1.5 m which not true). The authors' answer do not convince me that a global approach can be really work. We cannot change data cleaning, filtering and calculation methods in order to change the appearance of results.

(Remarks on code availability)

Version 5:

Reviewer comments:

Referee #1

(Remarks to the Author)

We are satisfied with the authors' responses. Based on their explanation of the analysis conducted, the data used, the figures shown, and importantly, the caveats and errors outlined alongside the dataset offered (which they have improved upon), we feel that it warrants publication.

(Remarks on code availability)

I downloaded the code. However, I do not have python set up on my computer so I couldn't read through it.

Referee #2

(Remarks to the Author)

I co-reviewed this manuscript with one of the reviewers who provided the listed reports.

(Remarks on code availability)

I am not fluent in python. I can only verify that the archive exists.

Referee #4

(Remarks to the Author)

Dear Editor,

We have completed our assessment of the revised manuscript and the authors' rebuttal. As requested, we focused primarily on the concerns raised by Reviewer 3. Overall, we find that the authors have carefully and thoroughly addressed all substantive points. In our view, their responses are scientifically sound and well supported, and the revised manuscript adequately resolves the issues raised during review. On this basis, we believe the manuscript is suitable for publication.

Below, we provide comments on several of the key points raised by Reviewer 3.

Validation of SWOT-derived tidal estimates - Reviewer 3 noted that the validation of SWOT-derived tidal estimates is insufficiently addressed. We respectfully take a different view. While SWOT observations are indeed relatively new, a substantial body of work has already been published on their characteristics and performance. The manuscript relies on the best-available SWOT products and builds on prior validation efforts documented in the literature. In our assessment, this constitutes a reasonable and well-justified methodological foundation.

Uncertainty of tidal amplitude estimates versus WSE errors - Reviewer 3 expressed skepticism regarding tidal amplitude errors of only a few centimeters, given reported water surface elevation (WSE) uncertainties on the order of 10–18 cm (68th percentile). We believe this concern arises from a comparison of quantities that are not directly equivalent. Tidal amplitudes are derived through harmonic analysis applied to extended WSE time series, and it is therefore expected that the uncertainty of fitted parameters can be substantially lower than that of individual observations. In our view, this difference does not indicate a methodological flaw.

Comparability with tide gauge observations - Reviewer 3 suggested that hourly-sampled tide gauge data may not capture the same dynamics as SWOT observations. However, the manuscript focuses on tidal amplitudes and phases derived from harmonic analysis of complete time series, rather than on point-by-point WSE comparisons. Tides have periods of several hours. Therefore, the time sampling interval only matters to correctly identify alias frequencies, but does not impact the comparability of the derived amplitudes.

Concerns related to SAR imaging geometry and dark water - Reviewer 3 raised the concern that water appears dark in a typical SAR imaging geometry, from which they derived the need for further data analysis / case studies. All results derived from the SWOT data are doubted hereafter. However, in contrast to typical SAR imaging systems like Sentinel-1, the looking angle of SWOT's radar antennae is tuned to 1–4 degrees (instead of ~30 degrees) in order to observe bright reflections from water instead. The authors demonstrate this in an illustrative plot, and using the statistics of the dark water flag, which is native to the data products. Brightness (and related SNR) can easily be assessed through the observations themselves, so that only little doubt can remain regarding the correctness of, e.g., the dark water flag. The authors defend their point well, by hinting to the relevant literature and documentation. This near-nadir geometry is one of the features that makes SWOT observations unique, so the point that the reviewer made here was completely unsubstantiated.

Use of median statistics - Finally, Reviewer 3 questioned the interpretation of median error values, suggesting that they obscure information about data spread and physical interpretation. While we agree in general that a single statistic cannot fully characterize error behavior, we note that—as the authors rightfully hint to—Fig. 2 shows all available data points in addition to the median errors. Both the scatter plots and the median errors are consistent, so we tend to object the notion of the reviewer in the context of this work.

(Remarks on code availability)

Referee #5

(Remarks to the Author)

I co-reviewed this manuscript with one of the reviewers who provided the listed reports.

(Remarks on code availability)

Referees' comments:

Referee #1 (Remarks to the Author):

This paper presents a first global atlas of observed tidal rivers, based on SWOT measurements. It links to a wonderful interactive web visualization of the identified rivers and associated metrics including tidal extent, amplitude, probability, and number of observations per node. Using the dataset, the authors examine the relationship between river characteristics and their tidal extent. This paper provides a world-first resource, and as such, an important and original contribution worthy of publication. Mapping the tidal characteristics of rivers globally, including regions remote or inaccessible, provides a rich resource for scientific exploration and discovery, and we expect it to be of interest to a wide-ranging readership. The paper is clearly written and well-presented (with a caveat regarding figures 3 and 4, see below). However, before it can be published, the authors should present a more detailed examination of the error budget. Second, the paper should provide more scientific insight – either directly or in the discussion of how the dataset can be utilized. Finally, the visualization of the data is uneven and should be improved upon to be suitable for a Nature publication.

A final comment – the Zenodo links provided for the dataset and the code were broken. We tried to search for them manually on Zenodo, but it appears they are not available, so we are unable to review these parts of the paper. This needs to be corrected in a revised edition.

We would like to thank these two reviewers for the constructive feedback and positive response to our manuscript. We have corrected the Zenodo link and provided more detailed responses to specific comments below.

Major points:

1. Error budget:

- Whilst exploring the web visualization, we noticed certain regions with abnormally large tidal amplitude far inland (e.g. east of Portland, OR). This appears to be an error but is labelled as 'likely tide'. If these sections of large amplitudes are not physical, can you try further quality control to remove the erroneous nodes?

We understand the reviewer's question and have made a major adjustment to clarify these "likely tide" classifications. Specifically, we have adjusted the quality control to use the recommendations by the SWOT community and we have applied an additional outlier detection method which can also help remove outliers in the amplitude estimations. In the final visualization of the data, we only incorporate amplitude estimations that exceed the uncertainty estimations made. This was taken into account for the classification but not in the visualization on the website and within the manuscript. We have added text to lines 217 - 236 of the revised manuscript.

- UTide is a robust analysis tool, well-suited for this kind of global survey; however, did you investigate using the NS_TIDE package for rivers where there is adequate discharge data to compare to results for UTide? (Matte et al. 2013, <https://doi.org/10.1175/JTECH-D-12-00016.1>) NS_TIDE is "adapted to the study of nonstationary signals, in this case, river tides. It builds the nonstationary forcing directly into the tidal basis functions. It is implemented by modification of T_TIDE; however, certain concepts, particularly the meaning of a constituent and the Rayleigh criterion, are redefined to account for the smearing effects on the tidal spectral lines by nontidal energy." If UTide tidal amplitudes are similar to NS_TIDE for well-constrained rivers, it would bolster global results for rivers where there are insufficient or non-existent discharge records. If there is substantial disagreement, this could help indicate how much error is introduced by neglecting the interaction of tides and river discharge.

We have had several discussions with the author of NS_TIDE, Pascal Matte. In principle, this would be an interesting insight to gain, but as we focus on a global analysis, this is not possible to do because the required high-quality discharge data are not available in all regions. As confirmed by Pascal, without the discharge data which we don't have for this study, the NS_TIDE software is exactly the same as UTide [which is also an extension of T-Tide]. This is an avenue of research that would be opened on the publication of this manuscript as a follow-up, where one could focus on refining regions based on specific wishes, but we believe it is beyond the scope of this manuscript. Doing this analysis on a global scale requires a significantly longer time series of observations for the tidal and discharge calculations, and is definitely an

interesting and highly impactful result that will be a suitable follow-up for the findings of this manuscript. While in principle it may be possible to use discharge data derived from SWOT itself for this purpose, no global SWOT-based discharge dataset has yet been published, and significant efforts to improve SWOT-based discharge algorithms are ongoing (Andreadis et al, 2025). As the SWOT discharge time series increases in length and fidelity, it will help enable the kind of analysis envisioned by the reviewer.

- Certain rivers are covered by the CalVal orbit and so have the benefit of daily repeat measurements. It would be interesting to quantify exactly how much more accurately the CalVal-covered rivers are mapped than the science orbit-only rivers. It would give a sense of error for the science orbit-only rivers.

We have done this analysis and added it to the supplementary Figure S2 text. We decided to not include it in the main text as we feel that regular readers of this manuscript will not be that interested and rather confused by this section. However, we agree scientifically this is valuable to keep for readers whom would be interested to find. We have added “SWOT has orbited in both a Cal/Val and science orbit, with some RiverSP nodes containing data from both orbits of the satellite. The Cal/Val phase of the satellite is particularly valuable as it allowed the retrieval of tidal constituents using very few observations based on tidal aliasing with relatively high levels of accuracy Hart-Davis et al, 2024). Taking this into account, the median amplitude error in reaches that contain both Cal/Val and science orbit observations is 4.85 cm and 3.62 cm, for M_2 and O_1 , respectively. These errors are 5.71 cm and 4.05 cm when using only the science orbit. These differences between the median amplitude errors indicate that, despite a combined Cal/Val and science orbit providing more observations, the results derived from RiverSP data from the science orbit alone are comparable to those combined with the Cal/Val. An important caveat in that statement is that the Cal/Val phase does not provide a global perspective and only very few suitable river tide reaches with in-situ measurements are covered, 62 tide gauges, compared to the full dataset, 754 tide gauges.”

- Currently, you validate the remote SWOT data against in-situ tide gauges via a simple spatial comparison (Figure 1). This is a nice figure, but we expected to see a more robust validation, given the thrust of this paper is the dataset resource itself. Can you include Extended Data figure(s) showing a scatter of all tide gauge amplitudes vs. SWOT estimates. Also, you should present the locations of the tide gauges along river, because, for example, if all tide gauges are within a few kilometers of the river mouth, that will bias the validation results compared to, say gauges located O(100km) upstream, where amplitudes are much smaller and SWOT estimates likely more error prone.

We have added two figures to demonstrate the validation of this dataset. We have also worked to improve the validation datasets by: 1) including a new dataset ArcTiCA (Hart-Davis et al 2024), which provides more coverage in the coastal Arctic regions, allowing for validation of some of these river tides; 2) producing a new global dataset, TICON-4, based on updates made to in-situ measurements in terms of quantity and quality. This has resulted in over 100 more tide gauges but most importantly, a significant improvement in the quality of the dataset that we now use for our validation (please see new Figure X in the main text). Additionally, we have added a supplementary figure, Figure S2, that demonstrates the errors binned as a function of distance to river mouth, number of gauges as well as a scatter plot comparing in-situ vs satellite derived amplitude estimations for both constituents. We have completely rewritten the section “Observing river tides from space” to account for these changes and discuss this in the “Validation datasets”.

- A time series comparison between river gauge data, SWOT data, and your estimated tidal components could be helpful to demonstrate how much non-tidal variability exists and how well SWOT matches in situ data; e.g. one case for different tidal probability regimes.

We have added a figure (Figure S3) showing four example validation gauges, one in a region where we have cal/val data influencing the data and discussed in lines XYZ. The top panel shows a gauge covered only by the science orbit and defined as tidal, the 2nd panel is also tidal and includes data from both the science and cal/val orbits, the 3rd shows a likely tidal region covered by both orbits, and the 4th is behind an obstacle and classified as non-tidal. We have added to lines 80 - 89:

“A direct comparison between the reconstructed water levels from SWOT against in situ observations from these three rivers further illustrates the fidelity of the SWOT analysis (Figure S3). We selected three gauges and associated SWOT time series where we have a combination of Cal/Val and science orbit data as well as profiles that contain classifications of ‘tide’, ‘likely tide’, and ‘no tide’. Using the appropriate tidal amplitudes and phases, we derived an estimation of tidal height and compared it against the tide gauge and raw RiverSP time series. Two of the three profiles, identified as ‘tidal’, reasonably match all three time series, indicating a clear dominance of the ocean tides in

these regions. In the 'likely tide' region, the tidal signal is still observable, but this time series is more influenced by nontidal effects. In the 'no tide' series, no significant tidal variability is observed or calculated from our dataset. This comparison exemplifies the importance of the new SWOT dataset for estimating tidal heights, which can be used in combination with ground-based data to help understand the role of tides on local river processes or to correct time series to study nontidal processes. ”

- You referenced studies, but it would be good to provide a concise explanation of the RiverSP error budget in the Method – i.e. what errors contribute to the error magnitude of 10-13 cm.

The expected SWOT error budget prelaunch is described in detail in Mission Error Budget Document as well as in the SWOT user's handbook, and is quite complex.

In summary, expected errors stem from a combination of random and systematic effects. The random effects are largely related to instrument thermal noise, while the systematic effects cover a large number of different areas including errors in the correction of effects from the wet troposphere, dry troposphere, and ionosphere; residual roll errors corrected using calibration at ocean cross-overs; errors associated with misclassification of land areas as water, which can result in the inclusion of higher land heights in node WSE calculations; the influence of specular reflections (often termed dark water), which can reduce data coverage and accuracy; specular ringing from strong signals at nadir, which can affect height errors in areas relatively near nadir; and a number of other potential error sources. Unfortunately, disentangling the influences of these different error sources on overall SWOT uncertainty is extremely difficult and has not yet been accomplished for inland waters. However, results from initial validation studies that we cite suggest that overall SWOT errors are largely in line with prelaunch expectations. We have added the following text to the methods (lines 232-238), as requested:

68th percentile differences between in situ gauge data and SWOT water surface elevations at the node scale range from 10-18 cm⁽³³⁻³⁵⁾. Sources of error in the SWOT data include residuals from the wet troposphere, dry troposphere, and ionosphere corrections; residuals in roll error corrected using calibration at ocean cross-overs; errors associated with misclassification of land areas as water, which can result in the inclusion of higher land heights in node WSE calculations; the influence of specular reflections (often termed dark water), which can reduce data coverage and accuracy; specular ringing from strong signals at nadir, which can affect height errors in areas relatively near nadir; and a number of other potential error sources. Detailed discussion of SWOT error sources are available in the SWOT User's Handbook⁵⁸.

2. Scientific Insight.

- In the conclusions, you discuss briefly some areas that will benefit greatly from this new global dataset. It would strengthen the paper if you can give more specific examples of how the dataset can be used, and more examples: e.g. Can you discuss the predictive value of the dataset? e.g. erosion and sedimentation, long-term changes from SL rise etc.?

A section termed “Implications for tidal rivers” has been added to the manuscript that presents a couple of examples we have produced. Additionally, we have discussed further in the conclusion some potential examples in lines 189 - 198.

- Some other thoughts:

- o Can you say anything about river geometry (e.g. a metric of total curvature?) vs. tidal extent?

We have looked at this, and we agree this could be an interesting study. Below is the CDF with respect to sinuosity for our dataset. It illustrates that this is not a simple statistic to implement, and we probably require more knowledge on width and depth of the rivers to properly quantify the effects of geometry on tidal extent. Simple curvature doesn't tell the full story. In Figure 1 for example with the Seine, we can see a highly meandering river with extremely large tides for a very large portion. Intuitively, the meandering should cause a dissipation in tides, but not much changes in the tides in these meanders. We agree, this is an interesting topic but can only be explored on a regional scale or with a lot more work on a global scale.

o The paper discusses tidal amplification upriver – can we learn anything new about the characteristics of such rivers from the large number of tidal rivers now identified?

Indeed, this new dataset represents an exciting tool for studying tidal amplification. However, we did not attempt to do any global-scale analysis of where and when tidal amplification occurs (for example, examine criteria or thresholds), which would require additional information on river depth and other geometric parameters. As the quality of these global datasets increases and as the timescale of the SWOT dataset continues to grow, we expect that many more studies of tidal amplification will be possible.

o Given what you have learned about river characteristics and tidal extents, can you say anything about the thousands of rivers too small to be measured by SWOT but which may be tidally influenced?

In Figure S4 and Lines 132 - 140, we have attempted to make a predictive formula on tidal extent across river scales. As more ground-based sensor data become available for coastal systems, it would be interesting to use ground-based studies of small tidal creeks to try to extend the predictive formula. At the moment, the standardized ground based datasets available from sources like the USGS are not focused on rivers below the scale of SWOT.

o Is there anything you can say about the characteristics of rivers that don't have tides?

We would suggest that our predictive analysis of tidal extent offers some insights on this matter, as it can be used to think about the portions of rivers that are no longer tidally influenced. We would also suggest that our analysis of the proportion of rivers where tides are obstructed, in Lines 141 - 152 and figure moved to supplement S5, gives us some insight into the role of humans in modifying river landscapes with dams and other structures that make tidal rivers non-tidal.

3. Visualization of data

• Figure 3 is important because it presents initial insights from the new dataset. However, it could be a more impactful visualization of those insights – it took work to understand what information is being shown and disentangle the different variables and their relationships. Is a cumulative distribution function the best way to present the results? A scatterplot would directly show the relationships between river extent and each of the variables. If you have strong objections to a different type of graph, improving this figure would help – use better contrasting colors for the different categories, and don't cumulatively bin the data in addition to the cumulative y-axis (use discrete bins for the variables in panels b,c,d, as you did in panel a).

The selection of colours and potentially using different graphs is challenging and was challenging in most figures. However, we have decided to do the colors scales as we discuss and show throughout to demonstrate how one statistic flows into another. I.e. the dark reds are closely aligned to the oranges and on and on. We have changed the discrete bins of the plots for Figure 4 at the reviewer's request.

• Figure 4 does not appear to add anything new (the same histogram could be plotted from the prior datasets, since the Strahler order and obstructions are not new to this study) and could be moved to Extended Data. What is new is that you can measure the tidal amplitude change due to obstructions. Perhaps there's a better way to visualize that? Unless all obstructions reduce the tidal amplitude to zero and don't impact downstream variability.

We agree. We have removed Figure 4 from the main text and added it to Figure S5, but kept the information within the text in lines 141 - 152.

• Considering multivariate data visualization, there are other types of graphs that could be employed for Figures 3 and 4 that may prove more fruitful, from scatterplots, box and whisker plots, parallel coordinates plots, radar plots, etc.

We tried several plots including all those mentioned by the reviewer, but they did not reveal anything new and were hard to read. We feel the selected plotting type for Figure 3 best demonstrates what we are discussing in the text and is the most intuitive of the approaches that we tested in our analysis.

Minor points (general):

• Explain all specific terms when you first mention them (SWOT RiverSP product, river slope, Strahler order, Pfafstetter Level 6, etc.)

We have added short definitions, explanations, or background references throughout the text.

• A suggestion for the introductory paragraph following the abstract: Nature's wide readership might appreciate a short visceral description of what happens as a tide floods and ebbs in the river – describing the process as it unfolds and including in the discussion how the flooding water (for example) impacts water quality, or how the ebb impacts erosion etc. (i.e. make the discussion of tidal river importance more cinematic and compelling)

We appreciated this comment and carefully considered ways to explain the dynamic processes that occur over a tidal cycle. We found it very difficult to generalize though, as the changes in sediment transport depend on where you are in the tidal reach (for example, organic sediment transport is heavily dictated by the saltwater-freshwater mixing zone). Similarly, changes in water quality depend on location (tidal freshwater zone versus brackish zone), channel geometry and whether mixing zones are stratified, and sources of contaminants (land or sea). Thus, we ultimately decided not to generalize, for fear of oversimplifying the unique nature of various tidal rivers and their longitudinal processes.

Minor points (by line number):

• Line 16: how do tides connect rivers with their surrounding floodplains? Can you be more direct, do you mean something like: "tides impact flooding of rivers and thus influence their floodplain extent"?

Amended with your recommendation.

• Line 36: expand to 'launched in December 2022'.

Amended.

• Line 49: in this section you should add the time period over which the study was conducted (can move or repeat from Methods section)

Done.

• Line 76: I would rephrase to something like: 'Worldwide, we identify [count] tidal rivers and more than 150,000 km of river extent that contain the pulse of the tide'

We changed this to: "Worldwide, we studied 51,627 river branches and identified more than 150,000 km of river extent that contain the pulse of the tide." Providing one number of tidal rivers from the SWORD database is not simple due to the number of connected branches within individual rivers which were analysed independently.

• Lines 79-83: Can you explain the discrepancy between in situ measurements showing a tidal amplitude of 2.3 cm near Obidios, and the fact that SWOT measurements also show tidal amplitudes (>10cm presumably) near there?

This sentence has changed based on the changes made to the analysis answered above and we no longer detect a tidal signal there anymore, note our threshold of greater than 10 cm. We have removed this.

• Line 92: Any surprises from un-gauged rivers measured for the first time by SWOT here?

In several regions, it is quite surprising to see how far the tides can propagate, for example, the Congo River that we discuss in the manuscript. The identification that we produce here, can open the door for future studies on these specific rivers to understand their influences on the local regions. In some rivers, it is also surprising how far the tides can propagate without any obstacles.

• Lines 95-96: Was this including classification 'likely tides', or only 'tides'?

We have clarified this, also relating to a comment below, by saying in lines 96 - 97: "The rivers are classified into 'tides', 'likely tides', 'likely no tides' and 'no tides' (see Methods)."

• Line 97: How did you estimate the amplitude of ocean tides at the mouth of the rivers? Was this EOT20 (Hart-Davis et al. 2021)?

This is done using SWOT RiverSP dataset that goes until the river mouth. We have added to that sentence in line 120: “ ocean tides at the mouth as defined by RiverSP”.

• Line 100: This is the first mention of ‘likely’ classification. I think it would be helpful for the reader who doesn’t want to get into the details of the Methods at the back of the paper to give a quick explanation of the categories tidal/likely tidal/likely non-tidal/non-tidal so that this paragraph makes sense on its own. Probably include it earlier at the beginning of this section or the previous section.

We have added a sentence explaining this at the beginning of the paragraph: “The rivers are classified into ‘tides’, ‘likely tides’, ‘likely no tides’ and ‘no tides’ (see Methods).”

• Line 167: give context to term ‘highest counter’:

Clarified to: “we prioritize files with the highest fidelity (reprocessing > forward processing) and the latest minor release version. ”.

• Line 214: How do tidal flats result in larger tidal amplitudes? My intuition tells me that tidal flats provide a “floor” on the approximate sinusoid of a tidal time series, which would bring the trough up and reduce the total amplitude. A one-sentence explanation or citation could clear this up.

This is correct. We have adjusted this ‘unrealistically larger tidal amplitudes’ to ‘unrealistic tidal amplitudes’.

• Line 238: give reference here for the ‘early study’.

Done

• Line 241: Can you provide the frequencies that M2 and O1 are aliased to, in both the cal/val and the science orbit periods (whilst stressing, as you have elsewhere, that it’s more complicated for each river node due to overlapping swaths).

Done and added to lines 251 - 253: “Additionally, this selection also accounts for restrictions based on the aliasing period of tidal constituents, which, in the worst case, when we ignore potential crossovers, would require 66 (12) and 53 (13) days of observations to produce an estimation of the M2 and O1 from the science phase (Cal/Val phase), respectively.”

Figures:

• Figure 1:

o In the legend the SWOT node is a black circle, but that is confusing since in the figure it is an unbroken line of color.

Ammended

o Why have you not plotted M2+O1 amplitude here as you do globally?

We have changed this.

o Can you plot the phase from both datasets? If so, it would be interesting to see, perhaps in Extended Data.

We have added it on the dahiti webpage and the data is within the dataset.

• Figure 2: Use a different color to separate ‘tide’ and ‘likely tide’ from the ‘likely no tide’ category.

We have kept these colourbars as the point is to demonstrate where there are tides, even when we classify them based on the uncertainty and classification we do in the manuscript.

• Figure S1: how do you arrive at the power law relationships plotted on the x axis? You need to fully explain.

We have updated the caption of this figure with the following text: “The relationship between tidal extent and readily measurable river parameters, including Strahler order at the mouth (O), river slope (S), and summed amplitude at the mouth (Am), derived from the global database. The scaling relationship on the x-axis was calculated from the individual correlations (r^2) between the logarithm of tidal extent and the logarithm of each of four individual variables: O, S, Am, and river width at the mouth. The latter was not included in the final scaling relationship due to low correlation.”

Referee #1 (Remarks on code availability):

The links to code and data were not working, so they could not be reviewed.

We apologize, as the public link was only in the cover letter to the editor. We have fixed the link in the manuscript too.

Referee #2 (Remarks to the Author):

I co-reviewed this manuscript with one of the reviewers who provided the listed reports.

Referee #3 (Remarks to the Author):

This manuscript is about investigating tides in coastal rivers on a global scale using satellite-based observations. The relevant scientific question authors like answering with this study is if the SWOT mission can be exploited to observe tides in 150,000 km of coastal rivers.

The estimation of the tidal field from space is not new in the open ocean and coastal zone. There is a coastal altimetry product coined X-TRACK Tidal Constant available at

<https://www.aviso.altimetry.fr/en/data/products/auxiliary-products/coastal-tide-xtrack.html>

This product provides along-track tidal constant estimates for 73 constituents using long time nadir altimetry missions (TOPEX/Poseidon + Jason series) within almost all coastal seas in the world. .

The novelty of this manuscript is the usage of the SWOT altimetry mission and the extension to the downstream part of rivers. The authors only estimate the diurnal O1 and semi-diurnal M2 tidal constituents, as they state the current SWOT record is too short. They use an high level product called "RiverSP" that provides preprocessed water surface levels spaced around 200 m. The estimation of tides is automatized in order to create a global atlas of tides within rivers. Validation of SWOT-derived tides is made using ground truth at 663 sites. Three rivers (Gironde, Seine, and Elbe) are analyzed in detail. Some other big rivers (Hudson, Amazon, Yangtze) are also highlighted.

Judging solely the presented results, the manuscript would be considered a ground-breaking in the field. However, I have some doubts that SWOT is capable of providing such stated stunning results. Overall the authors do not convince me. I see two important weaknesses: 1) the analyses of SWOT altimeter data is substantially absent considering that there are many caveats for this mission over inland waters; 2) the validation of SWOT-derived estimates of tides is poorly addressed.

We would like to thank the reviewer for their comments that have increased the validity of our findings and have resulted in the demonstrated value of our findings. As described above in response to the first reviewers, we have now presented a more expansive global validation. Previously, we had only included some brief numbers from a global analysis, but we agree that we needed to provide more information and have therefore added two new figures. Additionally, to justify the accuracy of the tides and the data being used to make the estimations, we produced a larger dataset for validation by adding over 100 more tide gauges. We also improved the overall quality of the validation dataset by taking the latest tide gauge data that has been corrected for errors that may influence the tidal estimation. We believe these changes to the way we present the validation have also made our results and findings more convincing. These can now be seen in the section: "Observing river tides from space" and in Figures 2, S1 and S2.

SWOT is essentially a SAR system looking down at very low incidence angles. It uses the interferometry principle to estimate surface level changes. The KaRIn instrument permits to acquire Single-Look-Complex images. Then a product called "PIXC" is derived that is the input for the "RiverSP" product used here. The algorithm to estimate surface level elevations is extremely complex. There is a static mask of prior water probability used during processing that informs where water might be expected. It is not perfect for two reasons: a) it cannot be perfect as water is dynamic; b) the transformation of ground coordinates to slant-plane coordinates is also uncertain and depends on the water height itself. Therefore, there is an impact in determining if a pixel is water or land or mixed.

The power of echoes is also important in the classification. The document titled "SWOT Science Data Products User Handbook (JPL D-109532, Revision A, March 2025) states that "water detection for HR data over land assumes that water is brighter than the surrounding land". An example of the Connecticut River is provided (page 169). If the water is not much more reflective than land, it may not be detected as water in the KaRIn ground processing. This would cause errors.

Other two important issues are: a) the frequent presence of smooth waters (known in SAR domain as dark waters) that reflect very little radar energy at the KaRIn off-nadir incidence angles; b) the brightness of side lobes (called specular ringing). There are other caveats (e.g., layover, phase unwrapping, etc.). Several quality flags are provided in the products and must to be carefully scrutinized.

The reviewer is correct that the method used to produce SWOT RiverSP data is complex, and there are indeed many potential sources of error. We have implemented more stringent data quality filtering using an

approach developed by the SWOT mission team at JPL (see lines 217-236). Ultimately, the proof of SWOT's accuracy is in its validation. It is true that there are substantial errors in some cases, but if appropriate data quality filtering is applied then characteristic one standard deviation (68th percentile) differences between in situ measurements and SWOT node data are on the order of 10-18 cm, depending on the river and the analysis performed. In the present study, we also perform additional smoothing, which likely decreases this node error further. Also note that some portion of this error comes from the inherent scale mismatch between the ~200 m spaced SWOT data and the in situ data, which even at the node scale can introduce additional apparent uncertainty. Even more pertinent is the validation of the actual tidal signals, which is now presented over a much larger number of gauges in Figure 2 and Figure S1 and S2. It is readily apparent that SWOT can provide an accurate measurement of tidal range in most cases. While in any global dataset there will inevitably be some areas that are less accurate than others, we are confident that our results reflect accurate tidal estimations and extents of influence in a large majority of cases.

Authors here just look at one flag if observations are “degraded” or “bad measurements”. It is important to highlight that “flags” alert you about potential issues that might affect the accuracy of the data. They simply inform users to look carefully those data. But also data flagged “good” might be “bad”. This is why a data analyses is necessary. SWOT is certainly an intriguing satellite mission, but the complex processing calls for a severe data screening and I don't see any analyses in the paper.

We thank the reviewer for this comment. Based on the community feedback and continued assessments on the SWOT data, we have improved the flagging of observations and implemented outlier detection. The selection of the flagging criteria improved our accuracy in the tidal estimations and the classification results. These are described in the methods section in lines 217-224.

There are published papers that already provide some indications that data analyses and validation is important. For instance Do Amaral et al. (2024) state “only about half of the WSE and WSS measurements meet the error budget and both require in situ validation for reliability. Additionally, we identified phase unwrapping and layover as significant sources of error at river locations near the nadir gap.”

F. Rodrigues do Amaral, T. Nguyen Trung, T. Pellarin and N. Gratiot, "Challenging SWOT: Early Assessment of Level 2 High-Rate River Products in an Urbanized, Low Elevation Coastal Zone," in IEEE Geoscience and Remote Sensing Letters, vol. 22, pp. 1-5, 2025, Art no. 1500605, doi: 10.1109/LGRS.2024.3501407.

It is absolutely true that data quality flagging is necessary in order to successfully use SWOT RiverSP data for an analysis such as the one presented here. It is indeed necessary to remove a significant fraction of the collected SWOT data in order to produce optimal results. This has been addressed in our previous response.

With regards to validation, the authors state that “Mean amplitude differences for the 663 sites were only 12.9 cm and 8.0 cm for the M2 and O1, respectively.” These values are of the same magnitude of the error of SWOT water surface elevations (10 to 13 cm). Statistics might not represent what happens at single sites. The performance is highly variable from site to site and from day to day. Validation is key step before exploiting a data set or scientific studies. Validation cannot be reduced to a single number.

We agree and have improved the representation of our validation throughout the manuscript. By adding additional data quality filtering, we have substantially reduced the mean amplitude differences to 11.5 cm for M2 and 6.4 cm for O1. Note, however, that we now choose to present the median values in the manuscript rather than the mean values, since a few significant outliers substantially affect the mean. As our goal is to provide characteristic tidal extents and ranges, we do not attempt to validate day-to-day variations in SWOT's performance. Indeed, it is not possible for SWOT to provide estimates of tides on daily timescales.

I looked at interactive map of Figure 2. I looked at the Mediterranean Sea that exhibits a micro-tidal regime. M2 variability is available from Figure 1 at

<https://os.copernicus.org/articles/20/1051/2024/#&gid=1&pid=1>

My eyes immediately noted the River crossing Rome (Italy). It shows M2+O1 amplitudes higher than 2 m that is clearly an artifact of the processing, as those values are not realistic in that area (see above figure).

A second example is the Ebro River (Spain). Ibañez et al. state that “The effect of tides is noticeable as far away as

Tortosa, 45 km from the mouth”. But the map indicates “likely no tide” at that place. Tidal amplitude at mouth is predicted around 8 cm. Mestres et al. show that the maximum astronomical tidal range in the area is about 25 cm, with an average value of 16 cm.

Ibañez, C., Pont, D., & Prat, N. (1997). Characterization of the Ebre and Rhone estuaries: A basis for defining and classifying salt-wedge estuaries. *Limnology and Oceanography*, 42(1), 89-101.

Mestres, M., Sierra, J. P., Sánchez-Arcilla, A., Del Río, J. G., Wolf, T., Rodríguez, A., & Ouillon, S. (2003). Modelling of the Ebro River plume. Validation with field observations. *Scientia Marina*, 67(4), 379-391.

A third example is the Po River (Italy). SWOT data predict “likely tides” with “50% probability” at Polesella (74 km from the mouth) with M2+O1 amplitudes around 60 cm. Moreover, the atlas predicts a tidal amplification along the river. But Nones et al. shows in Figure 2 tidal influence along the Po River and there are no tides at Polesella and no amplification along the river.

Nones, M., Maselli, V., & Varrani, A. (2020). Numerical Modeling of the Hydro-Morphodynamics of a Distributary Channel of the Po River Delta (Italy) during the Spring 2009 Flood Event. *Geosciences*, 10(6), 209. <https://doi.org/10.3390/geosciences10060209>.

The previous examples show questionable results probably related to artifacts of the processing (e.g., dark waters, geolocations errors, layover, ringing, etc.). However, a deeper examination of SWOT data is absent. The authors state “each river has unique morphologic and hydrologic conditions”. This means that the radar response is extreme variable. An in depth analyses of PIXC data, backscatter for each river, flags, radar scenario, etc. is necessary.

We would like to thank the reviewer for these comments. All these comments and errors are addressed based on the updated data flagging and outlier detections, as well as by taking into account the uncertainties in the webportal to only show amplitudes that exceed the uncertainty. Originally, we simply kept all the amplitudes in without taking into account the uncertainty information, which we did for the classification. Note we also are aware that we won't capture every river perfectly but we can confirm in this manuscript that we successfully implement a global approach that captures most tidal rivers properly.

It would not be possible to provide detailed analysis of SWOT PixC data, backscatter, and other variables for each river considered here, nor do we believe it would result in substantially improved results in most cases. Based on our validation results, we are confident that the combination of data quality filtering, outlier detection, and spatial smoothing used in this study results in sufficiently accurate data to allow the type of global-scale analysis described in the manuscript. However, based on a previous comment as well as this one we have discussed some issues and made appropriate references to suitable resources for readers in lines 225-236.

Also, the paper has major focus on a data set and some statistics. The reader would expect to see some new scientific insights from tidal fields in the rivers (e.g., tidal bore phenomena; tidal field propagation velocity; relationship with drought/flooding extreme events; salinity intrusions, etc.). But not a speculative statement like “Interestingly, within our global assessment, more than half of the investigated rivers (54.36%) exhibit e.g. upstream amplification similar to the Elbe”. If tides amplify as they travel upstream river it depends on local conditions. For instance I have doubt this happens e.g. if the river mouth is wide and open to the sea without much constriction or e.g. if there is high discharge. Anyway, the above statement must to be proved.

We agree that this single summary statistic was confusing how we phrased it. Now we have adjusted the way we derive the amplification and rephrased this statement. We suggest that tidal amplification is known to happen, and we have tried to highlight some specific examples where it is known to occur in Figure 1.

However, we feel that a global-scale summary statistic like the one we derive from this analysis is itself a novel finding that can only come from new datasets like this one. We have revised this sentence to:

“Interestingly, within our global assessment, the investigated rivers exhibit complex tidal characteristics like upstream amplification, which occurs in approximately 24.92% tidal rivers and is observed in the Elbe and the Gironde (Figure 1).”

Regarding new scientific insights, we agree that tidal bores are interesting. They have been observed with SWOT but do not require tidal amplitude analysis or tidal river classification to identify. The manuscript of Arildsen et al (2025) looked at this using SWOT KaRIn observations in the Bristol Channel, and we now reference this paper for the interested reader. Based on providing an impact or ‘some new scientific

insights', we have added an analysis on what we can learn about the relationship of tidal rivers with agriculture and an example in an extreme analysis in a tidal river. This is covered in "Implications of tidal extent in rivers".

A global atlas of tides from space in coastal rivers is certainly of interest to a wide audience, however, the analyses need to be rigorous. The reader expects that 1) SWOT observations are scrutinized in depth before estimating tides; 2) the tidal estimates are explained keeping in mind the relevant physical processes occurring in the selected sites.

The tidal classification data is available at Zenodo (<https://doi.org/10.5281/zenodo.15223861>)

The code to derive the classification as described within this manuscript is available at Zenodo (<https://doi.org/10.5281/zenodo.15223342>). However, both links do not work so I could not look at the data and code to check if paper's findings can be replicated.

An oversight on our part. We provided the direct links in the cover letter to the editor but failed to update the link in the manuscript itself. This is now fixed.

Statistics should be more robust, e.g., validation is poorly reported (only a mean value and unknown how water gauge minus altimetry varies from site to site). Figure S1 has data points widely scattered and error of correlation is not reported.

We have strengthened the validation. More details can be found in the earlier responses.

Referee #3 (Remarks on code availability):

I was not able to access data and code (probably there is a mistake in the link).

We have corrected the link and apologize for the inconvenience.

This email has been sent through the Springer Nature Manuscript Tracking System NY-610A-SN&MTS

Referee #1 and #2 (Remarks to the Author):

Summary:

The authors have made significant changes to the manuscript, which have improved its quality and addressed most of our concerns regarding the dataset's error budget, paper's scientific insight, and visualization of the data. We recommend this paper for publication in Nature. Below we have included our remaining concerns, which we believe would strengthen the paper for posterity.

Major Points

Using NS_tide

We agree that a comprehensive examination of rivers using NS_TIDE is infeasible and best saved for a later investigation; however, it is beneficial to compare a small number of well-constrained rivers—even just one—to see if the difference between UTide and NS_TIDE is enough to cause concern with the amplitude and/or error estimates. This is what we meant in our original comment. We understand that the uniqueness of each river means that no global conclusions should be drawn from checking a few select North American or European rivers, but the best river tide analysis tool out there is NS_TIDE, so regionally validating the global analysis with it would be insightful. If you have a strong and compelling objection to this, you should at least include an explanation of why NS_TIDE is not practical in the paper, e.g. in Methods -> Tidal Analysis, where UTide is introduced.

We have run some simulations with NS_Tide thanks to Pascal Matte, who is now included as a co-author in this manuscript. We agree that it is impractical to draw global conclusions based on these experiments, but we have added a brief discussion of a new analysis for two locations (from line 276):

“Extensions to classic harmonic analysis have been proposed in previous studies [Matte et al 2013] to account for the effects of time-varying external forcings on non-stationary stage and tidal components (termed NS_TIDE), which can influence tidal predictions in rivers. Preliminary comparisons in a dammed and unobstructed river between NS_Tide and UTide indicate good agreement in time-averaged tidal amplitude and extent estimates, with relative amplitude differences of 3.4% and 6.1% for the Seine (dammed) and Garonne (unobstructed) rivers. However, as SWOT time series lengthen and key products like river discharge continue to mature, extending the present analysis to explicitly resolve non-stationary tidal components will be a suitable follow-up. Such an extension will enable a more comprehensive assessment of how non-stationary processes modulate tidal amplitudes and extents in rivers at the global scale.”

To provide further detail on the new analysis, we ran NS_Tide for the Seine and Garonne rivers, to assess the differences in a dammed river versus not. In our experiments, we used upstream water levels as a proxy for river discharge (following the work described in Innocenti et al. (in review)). For the Seine and Garonne rivers, the match between HA amplitudes and median or mean (i.e., time-averaged) amplitudes from NS_Tide is good (relative differences of on average 3.4% and 6.1% for the Seine and Garonne (unobstructed), respectively). The median amplitude profiles and tidal extent agree well; however, a contrast emerges between obstructed and

non-obstructed rivers when employing non-stationary analysis. In the former case, the changes in tidal extent as a function of upstream “discharge” are minimal due to the river being tidal all the way up to the dam. In contrast, they are more pronounced in the latter case: the lower and higher amplitude quantiles from NS_Tide show that tides extinguish earlier for higher upstream water levels or river flows (around km 80 for Q10 in the Garonne) and farther upstream for lower flows (around km 95 for Q90). Because we used upstream water levels as an external forcing, it doesn’t tell us how much discharge creates this response, but it shows that there is a seasonality in amplitude and extent. As the SWOT orbit continues to produce data, this seasonality will be interesting to study and quantify on a global scale. However, for our analysis, the two methods are in agreement on average.

These findings not only lend confidence to our employed methods for deriving the tidal extent, but they also provide motivation for future work on this topic. The NS_Tide approach is attractive, but currently there are still methodological challenges to apply it globally (a longer time series is needed, large uncertainty in SWOT discharges, the selection of representative covariates in the presence of multiple tidal tributaries, dams, etc.).

Reviewer 3’s concerns on errors

We note that the website visualization still shows the river in Rome, Italy, with amplitudes > 3m, which apparently is unrealistic based on the reference provided by Reviewer 3.

We thank all three reviewers for this comment. On trying to identify why this happens in this case and the case in Algeria raised by the Reviewer 3, we identified two solutions that have been implemented. We had previously done the outlier detection at the reach scale, i.e. compared node estimation with all the nodes within a reach. We now do this outlier detection at the node scale, i.e. on each individual time-series. This has helped account for small-scale variations that had previously resulted in erroneous amplitude estimations. We have rewritten lines 239-241 to make this clear: *“To account for this, we implemented an outlier detection algorithm for **each node** to find and remove observations from the time series based on a robust 3-sigma test using the median absolute difference.”*

Additionally, in the online portal the plots of the amplitudes had previously also included regions which we have classified as ‘likely no tide’ or ‘likely tide’, i.e. our classification suggests that water level variations in these regions are not driven by tide. Actually, we would prefer not to provide these amplitude estimations, as without expert knowledge and the full dataset, the interpretation of these amplitudes would be rather challenging. We feel providing only the amplitude estimations when we predict ‘is tide’ is more suitable, so we have adjusted the web portal to only show amplitudes of M_2 and O_1 in regions we have confidently classified as ‘tide’. In these cases (in Italy and Algeria), we had classified this regions as ‘likely tide’ or ‘likely no tide’, i.e. we were not certain they were tidal, so we would have flagged these cases already.

Figures in main manuscript

Each figure of the main manuscript should be impactful, which we currently don’t feel is the case. Figure 2 is good but has limited information—Figure S2, which provides more clarity on errors, should replace it; Figure 2 would be best placed within Figure 3 as a multi-panel plot, as

it emphasizes the value of SWOT to move from very limited in situ observations to a truly global dataset. Finally, Figure 5 should be improved—it is not clear enough in its current format. A solution is to make it a multi-panel plot with zoom-in panels to better show the extreme high/low events. See minor points below for more comments on figures.

Figure S2 has now replaced Figure 2 and part of the caption has moved into the main text (lines 70-78). We have kept Figure 2 and 3 from the original manuscript (now Figure 3 and 4) separate as they are showing different results, and particularly Figure 4 (tidal extent) is the main outcome of this manuscript that we would like to stand on its own. We have added some zoom into some key events in Figure 5, highlighting the main talking points.

Minor Points

Title

The first line of your conclusion is "...to measure the tidal pulse of rivers from space...". Your title is "Observing the pulse of tidal rivers...". Our preference is to use the former in the title ('Observing the tidal pulse of rivers...') as it reads clearer for a general audience.

The title has been amended, as requested.

Figure 3

We disagree on the point of color palette. The greeny-blue of 'likely tide' and bluey-green of 'likely no tide' is very hard to visually distinguish on the global map, and incorrectly implies the tidal rivers are longer than they likely are. Use yellow or orange (or another suitably different color) and then there will be 3 distinct colors for the 3 distinct categories.

Amended.

Figure 4's line color choice:

Thank you for changing the bins, which we feel clarifies the results. It is indeed challenging to choose a good color scheme, but a "thermal" style colorbar like that used here is too difficult to distinguish between, especially for the 5-binned results in Figures 4b and 4d. A more distinct colormap such as "rainbow", "jet", or "turbo" will still possess an intuitive gradient but also better visualize the different lines.

Amended.

Figure S2

Plot M2 and O1 on different panels. In its current form it's difficult to distinguish the O1 data, it looks like the scatter shows little relationship between SWOT and tide gauge, but that might be because the density of points are not visible.

Amended.

Figure S3

Add y-axis values to show magnitudes.

Amended.

Aliasing periods of tides in SWOT data:

Thank you for stating the aliased periods for both cal/val and science orbits. However, it is phrased confusingly: in Zaron (2024) <<https://doi.org/10.1029/2024EA003677>>, which only focuses on the cal/val orbit, it is more clearly stated that 12 (M2) and 13 (O1) are the "Tidal Alias Periods" in days. As it is phrased in your paper, there is an ambiguity between observational

window and observation count (i.e., does the M2 need 66 snapshots from separate days from the science orbit to be observed, or just 4 flyovers that span a period longer than 66 days? I know that it's the latter, but it took a little bit to parse).

We have amended the text as follows:

*“Additionally, this selection also accounts for restrictions based on the aliasing period of tidal constituents, which, in the worst case, when we ignore potential crossovers, **would require observations spanning 66 (12) and 53 (13) days to produce an estimation of the M_2 and O_1 from the science phase (Cal/Val phase), respectively, as well as a span of 266 days (262) to separate the constituents from one another.**”*

Referee #3 (Remarks to the Author):

This paper is a resubmission of a paper that underwent to a first round of review. The authors attempt to showcase the abilities of SWOT for estimating tides in coastal rivers on a global scale (175,000 river kilometers). Overall there is no fundamentally improvement following my important recommendations.

During the first round of review I highlighted that “the analyses of SWOT altimeter data is substantially absent considering that there are many caveats for this mission over inland waters”. The authors didn’t follow my recommendation with data analyses as I expected to see in the revised paper. Then I told them that “the validation of SWOT-derived estimates of tides is poorly addressed”. In response to this point, the authors added more tide gauges and corrected for some errors, however, without specifying the nature of those errors. The validation approach in the paper is based on quantity rather than quality. It is evident that the global validation cannot be homogenous at the land/sea boundary. Any site has its own characteristics in terms of water dynamics, morphology, placement of tide gauge, etc. All tide gauges are of opportunity (e.g., hourly measurements means no almost coincidence in time with satellite) and not placed for a validation of satellite-based observations (e.g., in protected areas, therefore, the two measuring systems not observing the same dynamics). The authors added some cal/val sites, but there is no site-specific analyses to assess the error budget. The global results (Figure 2, S1 and S2) blur local details, i.e., they hide local variations. The authors state that SWOT-derived water surface elevations have 68 percentile errors in the range 10-18 cm. This means 32% of values are larger than those ones. Then the estimates of the tidal amplitudes are stated much lower 3-5 cm. This as “median error” that means something averaged blurring the local variations.

The SWOT WSE data has been extensively validated by several research groups both in published articles (see Maubant et al., 2025, Patibar et al., 2025, and Dhote et al., 2025) and by members of the SWOT Science Team. We state this in lines 334-343, where we take into account that amplitudes should always exceed a 13 cm threshold in our tidal extent analysis, which is based on literature stating the errors of WSE estimations are around this range (Maubant et al., 2025, Patidar et al., 2025). It also reflects work that is in the final stage of preparation, led by coauthor T. Pavelsky, which uses the exact same SWOT data quality flags as this manuscript and uses bespoke data collected in the field specifically to validate SWOT in 81 river reaches on four continents (Figure a). We provide the key water surface elevation validation CDFs below and would be happy to share the entire manuscript with the editor. In summary, we agree with the reviewer that rigorous validation of SWOT data is important, but that work is being done in multiple papers that are published, in review, and in preparation. Most published validation papers are based on long-term water surface elevation gauges (equivalent to tide gauges), and they show statistically very similar results to the analysis in Figure a which is based on much more spatially continuous measurements, suggesting that the Reviewer’s concerns about the “opportunistic” location of tide gauges is unlikely to be a major factor in our analysis.

Figure a: statistics of differences between water surface elevation from SWOT and in situ measurements from in situ data collected specifically for validation of SWOT on 12 rivers worldwide. The pertinent comparison is the filtered data, which shows a difference of ~15 cm. Note that this error includes error sources from SWOT, from the field data, and from the inevitable scale mismatch between the SWOT data and the point-scale field data.

As requested in the original round of review, we applied further data flagging to account for specific errors that could occur in the observation technique of SWOT, see lines 218-234. This resulted in improved tidal estimations, as further errors and outliers were detected and removed from our analysis. Although it is correct that our tidal amplitude errors are significantly less than these errors on average, we are not blurring local variations, as we discuss (see lines 50-78), and we demonstrate how these errors vary globally in multiple results [Figure 2, 3, and S1]. It is not surprising that the errors in our derived amplitudes are smaller than the errors in individual SWOT observations, as we rely on dozens of SWOT observations.

Regarding the statement that “All tide gauges are of opportunity,” We feel that the reviewer is misunderstanding what was done in the validation. The harmonic analysis was done on tide gauges (from the varying sources) and from SWOT, so our comparison does not rely on the ‘coincidence of timing’ because in tidal research we are deriving harmonic constants that have a set frequency and produce a tidal amplitude and phase lag for that specific location. We are aware that some gauges may have local variations, but that doesn’t negate their usability in a research context as has been done in hundreds of tidal studies (see e.g. Stammer et al 2014; Lyard et al 2021; Hart-Davis et al 2021). We also discuss limitations of the distance the tide gauge should be from the RiverSP measurement, which we have restricted to a maximum of 100 meters [see lines 288 - 290].

Another important point I highlighted was the particular radar acquisition scenario in inland waters that is totally different from open ocean. SWOT is a SAR and all SAR images over land show that water is “dark”, i.e., not reflective. This is why land is always more reflective than water. Going near nadir does not eliminate the issue, as the radar is always looking off. SWOT can only measure if water is more reflective than land. Is this true for 175,000 river kilometers at any revisiting time ? the authors did not prove that. Waters are frequently smooth over rivers. The authors here use a high level product. They apply some flags (i.e. something objectively stated), then furtherly apply a customized editing. There is no understanding of what happens in the processing chain from radar measurements to extracted information (tide amplitude). Therefore, it is really difficult to comment the results. Only using errors per site the reader would have a more convincing error budget assessment (that would include not only the radar, but also orbits, corrections, etc.). The authors agree with the issues but then they state that the global analyses show accurate results. As previously stated, the global analyses blur the site differences. As the tidal data set is the output of the paper the analyses has to be rigorous and convincing rather than approximate.

“The authors here use a high level product.” This is true, we use a high-level product provided by the NASA and CNES SWOT science project. These products are a result of decades of community research and work, as well as significant efforts to ensure the quality of the resultant datasets (see the RiverSP Algorithm Theoretical Basis Document (JPL D-56413, 2025) and the SWOT Data User’s Guide (JPL D-109532, 2025) referenced below). In the context of this submission, these data are used for scientific analysis, and the aim of this manuscript is not to itself provide a validation of each step of the product development, as this falls out of the scope of a *Nature* publication. Indeed, other publications have already demonstrated that SWOT measurements are sufficiently bright over rivers to provide highly usable water surface elevation data. For example, Dhote et al. (2025) finds a mean brightness of +13.34 dB over the Ganges River, which is within the prelaunch assumptions of +10-15 dB brightness over open water. While the reviewer is correct that near nadir incidence angles do not completely remove specular reflection from the water surface (which we term dark water), it is clear (as detailed below) that dark water is a problem in only a relatively small minority of cases. We agree that it is important to flag this and other sources of error, which we do according to best practices developed by the SWOT Science Team.

The reviewer appears to have either missed the discussions on the flagging procedures used or misunderstood them. The data quality flagging that is applied to this manuscript is emphasised in lines 218-232 of our manuscript. This is not subjectively chosen, but rather recommended by the SWOT RiverSP product developers. This adjustment was made in response to the Reviewers’ first comments, which has of course positively influenced our results compared to our previous approach. To demonstrate the impact of different flags, we have added lines 235 - 237:

“The most significant outlier criterion is the climatological ice flag, removing 80.9 million (40%) of the 195.3 million obtained RiverSP observations. Another 20.2 million observations (10% of the data) are removed due to a dark water fraction exceeding 0.4, which typically occurs with a calm, mirror-like water surface. After outlier rejection, 72 million (37%) observations remain.”

Additional data flagging takes into account other potential sources of error in the into WSE data, including errors in SWOT roll error corrections (described using the `xover_cal_b` flag) and a wide range of other error sources that are encapsulated in the `node_q` and `node_q_b` flags, which are described in detail in the SWOT RiverSP Product Description Document (see especially Table 12 in Appendix C). The development and application of these data quality flags was based on expert understanding of different sources of SWOT error, and we dispute the reviewer's characterization that "There is no understanding of what happens in the processing chain from radar measurements to extracted information (tide amplitude)." Moreover, the application of an additional set of filters to specifically address the impact of WSE on tidal estimation errors is also not subjective and is aimed at removing any errors that could be caused by errors remaining in the datasets. Outlier filtering is a standard practice in many altimetric remote sensing applications (see, for example, work by Quilfen and Chapron (2021), Schwatke et al. (2015), and Fernandes et al. (2014)). To summarize, we are aware that SWOT data itself has errors described in the referenced literature, and that these errors influence any analysis using these data. However, we are confident that our processing techniques adequately account for these errors, enabling the overarching aim of this paper, which is to map the tidal extent globally in rivers.

Finally, the authors definitely do not convince me when they state "By adding additional data quality filtering, we have substantially reduced the mean amplitude differences to 11.5 cm for M2 and 6.4 cm for O1. Note, however, that we now choose to present the median values in the manuscript rather than the mean values, since a few significant outliers substantially affect the mean." The median is just one number and ignores how the rest of the data are spread. There is no physical explanation behind numbers. The median value loses detail about how errors behave across sites or conditions.

We provided both the median and mean in our response to the reviewer. We are able to of course keep the mean in the main text if that is what is being suggested here, and the spread of data validation error is already captured in Figure 2, Figure 3 and S1. To resolve this concern, we have kept both the median and mean in the text. We have changed lines 55-56:

"Median and mean amplitude differences for the 622 sites were only 5.53 cm and 9.23 cm for M₂ and 3.23 and 5.75 cm for O₁."

During the first round of review I highlighted some examples of unrealistic amplitudes. I hoped authors analyzed in depth to understand reasons for unrealistic values. Now authors state that there were "errors" without explaining why and that the new data flagging and outlier detections solve. Just a quick look at the new map showed me other inconsistencies (e.g., Oued El Kebir (Algeria) with M2 of around 1.5 m which not true). The authors' answer do not convince me that a global approach can be really work. We cannot change data cleaning, filtering and calculation methods in order to change the appearance of results.

We appreciate this comment, and we have identified one source of error in our calculations and one point that is likely generating confusion. The reviewer is right that there are river amplitudes that are incorrect, and we have identified the reason behind this. As expanded on in the response to Reviewer 1, we had inadvertently previously done the outlier detection at the reach scale, i.e., compared node estimation with all the nodes within a reach. The scale of outlier

detection has been adjusted to the node scale, which accounts for individual time-series errors in each node. This node scale filtering globally removes about 2.5% more of the total observations than the reach scale filtering. We have adjusted lines 239-241 as follows: “*To account for this, we implemented an outlier detection algorithm for each node to find and remove observations from the time series based on a robust 3-sigma test using the median absolute difference.*”

Figure b below shows the different results for the Oued El Kebir with reach scale (top) and node scale (bottom) filtering. Note that with the new filtering, the confidence interval exceeds the amplitude right at the river mouth, so that no section has 100% tide probability, i.e. we would at no stage classify this river as ‘tide’. As can be seen in the amplitudes of the plots below, we estimate tides that are of larger amplitudes than what is expected, but we would like to emphasise that this is flagged out in our detection, which is shown in the tide probability in the figure below (same also applies for the previous Po example highlighted by Reviewer 1). The node scale based filtering accounts for more errors in the WSE, which in turn results in lower amplitude estimations, which in this example are below the confidence interval of the tidal estimation.

To reiterate a comment to reviewer 1/2, a potential source of confusion is that in the web portal plots of amplitudes, we decided to include amplitudes throughout without taking into account our tidal extent determination. This, of course, means we provided amplitude estimations when we predict no tides, which causes these apparent errors (which likely result from non-tidal factors) to be present. To make this online tool less confusing, we have now decided to only show amplitudes in regions that we identify as ‘tide’.

Regarding changing the appearance of the results, this seems to be a misunderstanding of the reviewer. For the tidal estimations [amplitudes], we do not do any cleaning or filtering or changing of methods in the validation of the tidal estimations. Cleaning is only done on the RiverSP WSE data to remove potential erroneous measurements to improve our tidal estimations. The flagging procedures have been adjusted between versions of the manuscript in an attempt to deal with the justified concerns of the reviewer in the first response (Lines 218-234), which resulted in the removal of unreliable data [we calculate that with all flagging adjustments and node-scale filtering we currently lose about 65.5% of all RiverSP data within the rivers we study] and hence improved our resultant tidal estimations. The inclusion of more tide gauges, which themselves have been improved due to greater data availability, also results in the median statistics being improved. However, this does not mask that there are still some erroneous sites that have been kept in our results and can be seen in Figures 2,3 and S1. The tidal amplitude estimations are not themselves adjusted to change the appearance of the results. To further emphasise, the main aim of this manuscript is to produce tidal extent estimations. To achieve this goal, errors in WSE as well as tidal estimations are taken into account and influence the confidence of our extent estimations.

Figure b: Old reach scale filtering (top) and new node scale filtering (bottom) along the Oued El Kebir.

References

Dhote, P.R., Agarwal, A., Paris, A., Singhal, G., Thakur, P.K., Oubanas, H., Moreira, D.M., Gal, L., Garg, V., Kumar, P. and Singh, R.P., 2025. Unveiling the first impressions of the wide-swath altimetry SWOT mission over the Ganga River, India. *Geophysical Research Letters*, 52(19), p.e2025GL115402.

Fernandes, M. J., Lázaro, C., Nunes, A. L., & Scharroo, R. (2014). Atmospheric corrections for altimetry studies over inland water. *Remote Sensing*, 6(6), 4952-4997.

Quilfen, Y., & Chapron, B. (2021). On denoising satellite altimeter measurements for high-resolution geophysical signal analysis. *Advances in Space Research*, 68(2), 875-891.

Schwatke, C., Dettmering, D., Bosch, W., & Seitz, F. (2015). DAHITI—an innovative approach for estimating water level time series over inland waters using multi-mission satellite altimetry. *Hydrology and Earth System Sciences*, 19(10), 4345-4364.

JPL D-105505 (2023), SWOT RiverSP Algorithm Theoretical Basis Document [Accessed Jan 8, 2026]

https://archive.podaac.earthdata.nasa.gov/podaac-ops-cumulus-docs/web-misc/swot_mission_docs/atbd/D-105505_SWOT_ATBD_L2_HR_RiverSP_20230713_cite.pdf

JPL D-109532 (2025) SWOT Data User's Handbook [Accessed Jan 8, 2026]

https://www.earthdata.nasa.gov/s3fs-public/2024-06/D-109532_SWOT_UserHandbook_20240502.pdf

JPL D-56413 (2025) SWOT RiverSP Product Description Document [Accessed Jan 8, 2026]

https://archive.podaac.earthdata.nasa.gov/podaac-ops-cumulus-docs/web-misc/swot_mission_docs/pdd/D-56413_SWOT_Product_Description_L2_HR_RiverSP_20250224a_R evC_clean_sig_final.pdf